# TOWARDS OPTIMAL FEATURE-SHAPING METHODS FOR OUT-OF-DISTRIBUTION DETECTION

**Qinyu Zhao[1], Ming Xu[1], Kartik Gupta[2], Akshay Asthana[2], Liang Zheng[1], Stephen Gould[1]**
[1] The Australian National University
[2] Seeing Machines Ltd
`{qinyu.zhao,mingda.xu,liang.zheng,stephen.gould}@anu.edu.au`
`{kartik.gupta,akshay.asthana}@seeingmachines.com`

## ABSTRACT

Feature shaping refers to a family of methods that exhibit state-of-the-art performance for out-of-distribution (OOD) detection. These approaches manipulate the feature representation, typically from the penultimate layer of a pre-trained deep learning model, so as to better differentiate between in-distribution (ID) and OOD samples. However, existing feature-shaping methods usually employ rules manually designed for specific model architectures and OOD datasets, which consequently limit their generalization ability. To address this gap, we first formulate an abstract optimization framework for studying feature-shaping methods. We then propose a concrete reduction of the framework with a simple piecewise constant shaping function and show that existing feature-shaping methods approximate the optimal solution to the concrete optimization problem. Further, assuming that OOD data is inaccessible, we propose a formulation that yields a closed-form solution for the piecewise constant shaping function, utilizing solely the ID data. Through extensive experiments, we show that the feature-shaping function optimized by our method improves the generalization ability of OOD detection across a large variety of datasets and model architectures. [1]

## 1 INTRODUCTION

Out-of-distribution (OOD) detection aims to identify test samples that fall outside the inherent training label space, given a deep learning model pre-trained on an in-distribution (ID) training set. To detect OOD samples, OOD scores, such as maximum softmax probability (MSP) (Hendrycks & Gimpel, 2016) and energy score (Liu et al., 2020) are computed using the logits estimated by the model, where a lower score indicates a higher probability that the sample is OOD.

Feature shaping (Sun et al., 2021; Djurisic et al., 2022; Xu & Lian, 2023; Song et al., 2022) refers to a family of methods that manipulate the underlying feature representations, typically from the penultimate layer of a pre-trained model, such that OOD scores can more effectively distinguish between ID and OOD data. Representative methods like ReAct (Sun et al., 2021) and ASH-P (Djurisic et al., 2022) employ element-wise feature clipping and percentile-based feature pruning, respectively.

Despite their notable achievements, existing feature-shaping methods usually rely on rules manually designed for specific OOD datasets and model architectures, which consequently limit their generalization ability. Our observations indicate that variations in OOD datasets and model architectures can cause significant performance degradation in these methods. For instance, when using transformer-based models, some approaches, despite achieving state-of-the-art results on established benchmarks, perform no better than random guessing. These observations emphasize the necessity for a comprehensive understanding and comparison of the current feature-shaping methods and the development of a more generalizable approach.

To this end, we first formulate an abstract optimization problem that encapsulates all feature-shaping OOD detection methods. In a nutshell, our formulation entails finding the shaping function that maximizes some performance objective for OOD detection.

---

[1]Our code is available at https://github.com/Qinyu-Allen-Zhao/OptFSOOD.

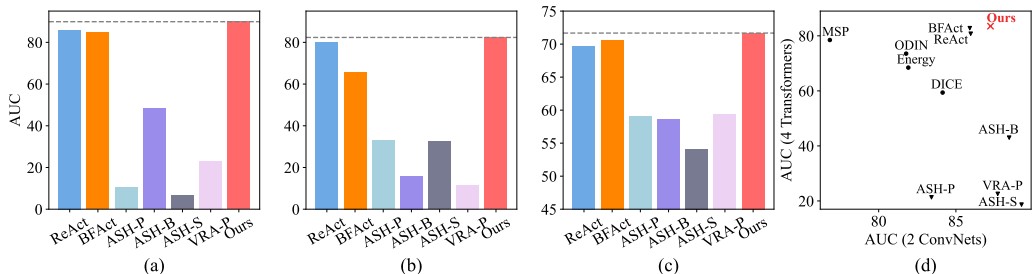

Figure 1: Comparing our method with existing feature-shaping methods. The dashed lines denote the performance of our method for comparison. (a) ImageNet (ID) *vs.* iNaturalist (OOD) with ViT-B-16; (b) ImageNet (ID) *vs.* iNaturalist (OOD) with MLP-Mixer-B; (c) CIFAR100 (ID) *vs.* CIFAR10 (OOD) with MLP-Mixer-Nano; (d) Average performance of different methods across eight OOD datasets with two ConvNets and with four transformer-based models.

To make the optimization problem concrete and tractable, we propose a piecewise constant shaping function and formulate an objective function that maximizes the expected difference between the maximum logit values for ID and OOD data, assuming the availability of OOD training samples. An important benefit of this formulation is that it highlights the operational mechanisms underpinning previous feature-shaping methods. Namely, the shaping functions used by existing feature-shaping methods including ReAct (Sun et al., 2021), BFAct (Kong & Li, 2023), and VRA-P (Xu & Lian, 2023), approximate the optimal shaping function found by our approach.

Using our framework, we further propose a feature-shaping method that can be optimized without access to OOD data. Specifically, through the analysis of the effect of OOD data on the optimal solution compared to ID data, we argue that it is feasible to remove the OOD-related term in the objective function, thereby deriving a closed-form solution based on ID data only. Empirical observations in experiments support our claim.

Our experiments compare our proposed approach to existing feature-shaping methods, utilizing a more comprehensive experimental setup than previous works. We use CIFAR10, CIFAR100 (Krizhevsky et al., 2009), and ImageNet-1k (Russakovsky et al., 2015) as ID datasets. For each ID dataset, we use eight OOD datasets for evaluation. Furthermore, we evaluate all methods using a wide variety of models, including convolutional networks (ConvNets), transformers and fully multi-layer perceptron (MLP) models, the latter two of which have not been extensively studied in prior works. As shown in Fig. 1, compared with other feature-shaping methods, ours demonstrates more generalized OOD detection performance across a broad spectrum of datasets and model architectures.

In summary, this paper covers three main points. First, we develop a general optimization formulation for feature-shaping methods and show that a concrete instance of this problem provides insight into the mechanisms of previous methods. Second, we propose a novel approach for training a generalizable feature-shaping method for OOD detection without access to OOD data. Last, our approach achieves more reliable OOD detection performance across various datasets and models.

## 2 A RECAP OF FEATURE SHAPING FOR OOD DETECTION

In this section, we define the problem of OOD detection for image classification, with particular emphasis on feature-shaping approaches, and subsequently, introduce our general optimization formulation of feature shaping for OOD detection.

### 2.1 OOD DETECTION FOR IMAGE CLASSIFICATION

Assume that we have an ID dataset, defined as a joint distribution $D_{\mathcal{X},\mathcal{Y}}^{\text{ID}}$ over input observations $\mathcal{X}$ and discrete labels $\mathcal{Y}$ for a given model $\mathcal{M}$. Furthermore, assume we have an OOD dataset $D_{\mathcal{X},\mathcal{Y}'}^{\text{OOD}}$. Previous works (Sun et al., 2021; Sun & Li, 2022; Yang et al., 2021) assume $\mathcal{Y} \cap \mathcal{Y}' = \emptyset$. Inherent in this assumption is that the marginal distributions over observations $\mathcal{X}$ for ID and OOD datasets do not coincide. That is, $\mathcal{D}_{\mathcal{X}}^{\text{ID}} \neq \mathcal{D}_{\mathcal{X}}^{\text{OOD}}$.

The model, usually a neural network, maps observations to logits, $\mathcal{M} : \mathcal{X} \to \mathbb{R}^C$, where $C = |\mathcal{Y}|$ denotes the number of classes. It is trained on a dataset $S_{\text{train}} = \{(\boldsymbol{x}^{(i)}, y^{(i)})\}_{i=1}^N$, comprised of $N$ i.i.d. samples drawn from $D_{\mathcal{X},\mathcal{Y}}^{\text{ID}}$. Furthermore, for typical architectures for image classification, we can decompose the model $\mathcal{M} = \mathcal{W} \circ \mathcal{B}$ into a backbone model, $\mathcal{B} : \mathcal{X} \to \mathbb{R}^M$, and linear classifier $\mathcal{W} : \mathbb{R}^M \to \mathbb{R}^C$. $\mathcal{B}$ extracts penultimate layer feature representation $\boldsymbol{z} \in \mathbb{R}^M$ of an input observation $\boldsymbol{x}$, and $\mathcal{W}$ is a linear layer that maps $\boldsymbol{z}$ to logits $\boldsymbol{\ell} \in \mathbb{R}^C$. Different from conventional machine learning practice, the test set $S_{\text{test}}$ contains inputs sampled from the joint distribution $D_{\mathcal{X}}^{\text{ID}} \cup D_{\mathcal{X}}^{\text{OOD}}$, where $D_{\mathcal{X}}^{\text{ID}}$ and $D_{\mathcal{X}}^{\text{OOD}}$ denote the marginal distributions of $D_{\mathcal{X},\mathcal{Y}}^{\text{ID}}$ and $D_{\mathcal{X},\mathcal{Y}'}^{\text{OOD}}$, respectively.

The goal of OOD detection is to design a score function $s$ to indicate whether a sample $\boldsymbol{x}$ is from $D_{\mathcal{X}}^{\text{ID}}$ or $D_{\mathcal{X}}^{\text{OOD}}$. MSP, maximum logit score (MLS), and energy score serve as examples of scores for OOD detection. They are defined based on logits $\boldsymbol{\ell}$ as:

$$s_{\text{MSP}}(\boldsymbol{\ell}) = \max_{i=1,\dots,C} \frac{\exp(\ell_i)}{\sum_{j=1}^C \exp(\ell_j)}, \quad s_{\text{MLS}}(\boldsymbol{\ell}) = \max_{i=1,\dots,C} \ell_i, \quad s_{\text{Energy}}(\boldsymbol{\ell}) = \log \sum_{j=1}^C \exp(\ell_j), \quad (1)$$

where $\ell_i$ is the logit for the $i$-th class. For the scores discussed above, a lower value indicates a higher likelihood of the sample being OOD. In practical applications, a threshold, denoted by $t$, is utilized to separate ID samples from OOD ones. If the score of a sample surpasses this threshold, the sample will be classified as ID, and OOD otherwise.

## 2.2 FEATURE-SHAPING

A feature-shaping method uses a shaping function $\boldsymbol{f} : \mathbb{R}^M \to \mathbb{R}^M$ to manipulate features, i.e., $\bar{\boldsymbol{z}} = \boldsymbol{f}(\boldsymbol{z})$. The goal of developing a feature shaping method for OOD detection can be formulated as solving the following optimization problem,

$$
\begin{aligned}
\text{maximize (over } \boldsymbol{f}) \quad & \mathbb{E}_{\boldsymbol{x}^{\text{ID}} \sim D_{\mathcal{X}}^{\text{ID}}, \boldsymbol{x}^{\text{OOD}} \sim D_{\mathcal{X}}^{\text{OOD}}} \left[ \mathcal{P} \left\{ s(\bar{\boldsymbol{\ell}}^{\text{ID}}), s(\bar{\boldsymbol{\ell}}^{\text{OOD}}) \right\} \right], \\
\text{subject to} \quad & \bar{\boldsymbol{\ell}}^{\text{ID}} = \mathcal{W}(\bar{\boldsymbol{z}}^{\text{ID}}), \quad \bar{\boldsymbol{z}}^{\text{ID}} = \boldsymbol{f}(\mathcal{B}(\boldsymbol{x}^{\text{ID}})), \\
& \bar{\boldsymbol{\ell}}^{\text{OOD}} = \mathcal{W}(\bar{\boldsymbol{z}}^{\text{OOD}}), \quad \bar{\boldsymbol{z}}^{\text{OOD}} = \boldsymbol{f}(\mathcal{B}(\boldsymbol{x}^{\text{OOD}})),
\end{aligned}
\quad (2)
$$

where $\mathcal{P}$ is a given performance measure. For example, a common measure is the area under the receiver operating characteristic curve (AUC), defined as $\mathcal{P}_{\text{AUC}}(s_1, s_2) = \mathbb{1}\{s_1 > s_2\}$. Existing shaping functions $\boldsymbol{f}$ can be broadly categorized into two types: (i) element-wise, such as ReAct (Sun et al., 2021) and VRA-P (Xu & Lian, 2023), which applies a global element-wise mapping to each individual feature, i.e., $\bar{z}_i = f(z_i)$, or (ii) whole-vector mappings such as ASH (Djurisic et al., 2022), which manipulate the full feature vector $\boldsymbol{z}$.

## 3 OPTIMAL FEATURE SHAPING

In this section, we propose a specific instance of the general formulation described in Section 2, following previous element-wise feature-shaping methods (Sun et al., 2021; Xu & Lian, 2023; Kong & Li, 2023). To optimize the shaping function, we initially examine which feature value ranges are effective for OOD detection. A significant challenge arises from the continuous nature of feature values. To address this, we partition the domain of features into a disjoint set of intervals, which allows us to approximate the optimal shaping function with a piece-wise constant function. With this approximation, we can formulate an optimization problem to maximize the expected difference between the maximum logit values of ID and OOD data. Subsequently, we show that the optimized shaping function using OOD samples bears a striking resemblance to prior methods, shedding light on the mechanics of these methods. Finally, we propose a formulation that does not require any OOD data to optimize and admits a simple, closed-form solution.

### 3.1 VALUE- AND INTERVAL-SPECIFIC FEATURE IMPACT

Given a sample $\boldsymbol{x}$, the maximum logit $\ell^{\max}$ (disregarding the bias term) is the dot product between the corresponding weight vector $\boldsymbol{w}^{\max}$ and its feature representation $\boldsymbol{z}$, specifically,

$$\ell^{\max}(\boldsymbol{z}) = \boldsymbol{w}^{\max} \cdot \boldsymbol{z} = \sum_{i=1}^M w_i^{\max} z_i. \quad (3)$$

Assuming an element-wise feature-shaping function $f : \mathbb{R} \to \mathbb{R}$, define $\theta(z_i) \triangleq \frac{f(z_i)}{z_i}$. Then assuming the weights $\boldsymbol{w}^{\max}$ do not change[2], the maximum logit after shaping is:

$$\bar{\ell}^{\max}(\bar{\boldsymbol{z}}) = \sum_{i=1}^{M} w_i^{\max} f(z_i) = \sum_{i=1}^{M} \theta(z_i) w_i^{\max} z_i. \tag{4}$$

To design $\theta(z)$ and understand which feature value ranges are effective for OOD detection, we compute the contribution of features with specific values to the maximum logit. Concretely, we define the value-specific feature impact (VSFI): $V(z) = \sum_{i=1}^{M} \mathbb{1}\{z_i = z\} w_i^{\max} z_i$. However, $z$ is a continuous variable, and real-world data only provide a countable number of features, which can gives overly sparse distributions of VSFIs. To address this, we partition the domain of individual features into a disjoint set of intervals $\mathcal{A}$ that cover $z_i$. Formally, the interval-specific feature impact (ISFI) on an interval $[a, b) \in \mathcal{A}$ is given by:

$$I_{[a,b)}(\boldsymbol{z}) = \sum_{i=1}^{M} \mathbb{1}\{a \le z_i < b\} w_i^{\max} z_i = \sum_{i:a \le z_i < b} w_i^{\max} z_i. \tag{5}$$

Then, for a single sample, the maximum logit is the summation over all ISFIs, given by:

$$\ell^{\max}(\boldsymbol{z}) = \sum_{[a,b) \in \mathcal{A}} I_{[a,b)}(\boldsymbol{z}). \tag{6}$$

In our experiments, we divide feature values into equal-width intervals. Given that feature distributions typically exhibit long-tailed characteristics (Sun et al., 2021), we select the 0.1 and 99.9 percentiles of all features in a training set as the lower $\alpha$ and upper $\beta$ limits of feature values, respectively. We then partition the range $[\alpha, \beta)$ into $K = 100$ equal-width intervals. This results in a set of intervals $\mathcal{A} = \{[a_k, b_k)\}_{k=1}^{K}$, where $a_k = \alpha + (k-1)\delta$, $b_k = \alpha + k\delta$, and $\delta = \frac{\beta - \alpha}{K}$ for $k = 1, \dots, K$. Then, the maximum logit can be expressed as:

$$\ell^{\max}(\boldsymbol{z}) = \sum_{[a,b) \in \mathcal{A}} I_{[a,b)}(\boldsymbol{z}) = \sum_{k=1}^{K} I_{[a_k,b_k)}(\boldsymbol{z}), \tag{7}$$

and for each data sample, we can generate a $K$-dim ISFI vector denoted as:

$$\boldsymbol{I}(\boldsymbol{z}) = \left( I_{[a_1,b_1)}(\boldsymbol{z}), \dots, I_{[a_K,b_K)}(\boldsymbol{z}) \right) \in \mathbb{R}^K. \tag{8}$$

## 3.2 OPTIMIZING THE RESHAPING FUNCTION

We now describe how to optimize $\theta(z)$, assuming $\theta(z)$ is piecewise constant over the intervals $\mathcal{A}$. For an interval $[a_k, b_k)$, we estimate a scalar $\hat{\theta}_k$ as a constant approximation to $\theta(z)$ within the interval. In the limit as the interval widths shrinks to zero, we recover the continuous, element-wise shaping function. We can approximate the maximum logit as,

$$\bar{\ell}^{\max}(\bar{\boldsymbol{z}}) = \sum_{k=1}^{K} \hat{\theta}_k I_{[a_k,b_k)}(\boldsymbol{z}). \tag{9}$$

While previous works (Sun et al., 2021; Xu & Lian, 2023) evaluate their methods by computing the expected difference in maximum logits between ID and OOD data, optimizing this criteria solely does not improve OOD detection performance. To see this, observe that simply uniformly scaling features by a large factor increases the expected difference in maximum logits without enhancing the differentiation between ID and OOD samples. This motivates introducing regularization on the shaping function and leads us to the following problem [3]:

$$\begin{aligned} \text{maximize (over } \boldsymbol{\theta}) \quad & \mathbf{E}_{\boldsymbol{x}^{\text{ID}} \sim D_{\mathcal{X}}^{\text{ID}}, \boldsymbol{x}^{\text{OOD}} \sim D_{\mathcal{X}}^{\text{OOD}}} \left[ \boldsymbol{\theta}^{\mathsf{T}} \boldsymbol{I}(\boldsymbol{z}^{\text{ID}}) - \boldsymbol{\theta}^{\mathsf{T}} \boldsymbol{I}(\boldsymbol{z}^{\text{OOD}}) \right] \\ \text{subject to} \quad & \|\boldsymbol{\theta}\| = \sqrt{K}, \quad \boldsymbol{z}^{\text{ID}} = \mathcal{B}(\boldsymbol{x}^{\text{ID}}), \quad \boldsymbol{z}^{\text{OOD}} = \mathcal{B}(\boldsymbol{x}^{\text{OOD}}), \end{aligned} \tag{10}$$

---

[2]We validate this assumption in Section B.2.

[3]We constrain $\|\boldsymbol{\theta}\|$ to be equal to $\|\mathbf{1}\| = \sqrt{K}$ instead of other values. Reasons can be found in Section A.2.

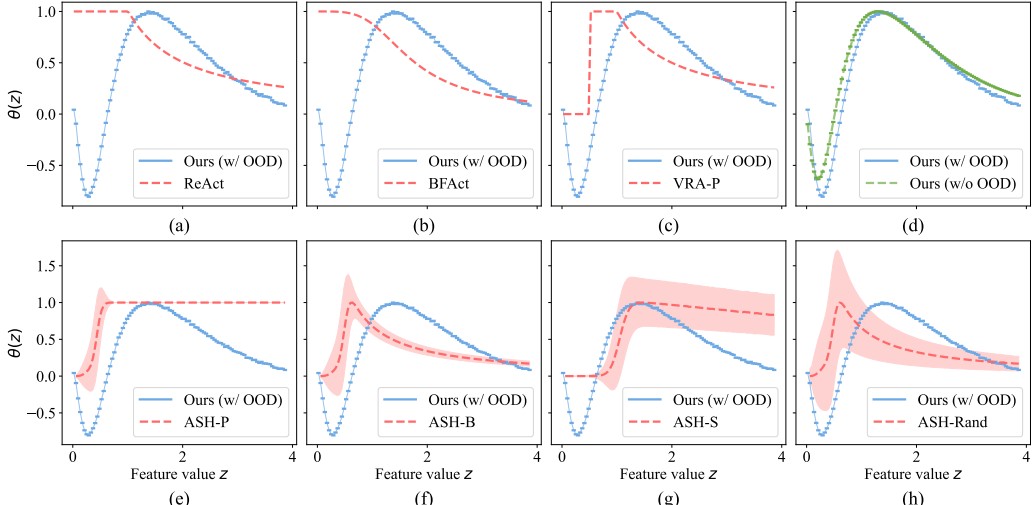

Figure 2: Visualization of shaping functions. The blue lines (ours w/ OOD) derive from Eq. 10, while the green line (ours w/o OOD) from Eq. 14. Red lines represent different existing methods, while shaded regions indicate estimated standard deviations. $\theta$ has been rescaled for the best visualization.

where due to the linearity of expectation, the objective function can be expressed as:

$$\boldsymbol{\theta}^{\mathsf{T}} \left( \mathbf{E}_{\boldsymbol{x}^{\mathrm{ID}} \sim D_{\mathcal{X}}^{\mathrm{ID}}} \left[ \boldsymbol{I}(\boldsymbol{z}^{\mathrm{ID}}) \right] - \mathbf{E}_{\boldsymbol{x}^{\mathrm{OOD}} \sim D_{\mathcal{X}}^{\mathrm{OOD}}} \left[ \boldsymbol{I}(\boldsymbol{z}^{\mathrm{OOD}}) \right] \right). \tag{11}$$

We extract ISFI vectors for the ImageNet-1k training set (ID) (Russakovsky et al., 2015) and for the iNaturalist dataset (OOD) (Van Horn et al., 2018), using a ResNet50 model pre-trained on ImageNet-1k training set. The estimated $\hat{\theta}_k$ is plotted in Fig. 2.

For completeness, we describe existing feature-shaping methods (Sun et al., 2021; Kong & Li, 2023; Xu & Lian, 2023) using our reparameterization of the feature shaping problem with $\theta(z)$,

$$\theta_{\mathrm{ReAct}}(z) = \frac{\min(z, t)}{z}, \quad \theta_{\mathrm{BFAct}}(z) = \frac{1}{\sqrt{1 + (\frac{z}{t})^{2N}}}, \quad \theta_{\mathrm{VRA\text{-}P}}(z) = \begin{cases} 0 & z < \text{low} \\ 1 & \text{low} \leq z < \text{high} \\ \frac{\text{high}}{z} & \text{high} \leq z, \end{cases}$$
$$\tag{12}$$

where $t$, $N$, low, and high are hyperparameters. We then plot the resultant $\theta(z)$ in Fig. 2(a-c). We show that these prior works are empirically approximating our optimal piecewise constant shaping function. A difference between our shaping function and VRA-P, is that VRA-P sets low-value features to zero while ours flips the sign of these features to better utilize them for OOD detection.

Then, we extend our analysis to other feature-shaping methods. ASH-P, -B, -S, and -Rand denote four simplistic yet effective methods introduced by Djurisic et al. (2022). In contrast to ReAct, which employs a global threshold to clip features, ASH methods incorporate the entire feature vector and utilize a local threshold to process the feature representation of each instance. Their shaping functions are not reducible to element-wise functions.

We instead approximate the average shaping effect within each feature value interval. Specifically, we utilize ImageNet-1k validation set (ID) and iNaturalist (OOD) with ResNet50. For an interval $[a_k, b_k)$, we retrieve all features from ID and OOD datasets that fall within this range, along with their reshaped counterparts. Following this, the ratio of reshaped and original values are computed as $\theta_k$ for a specific data instance. Subsequently, we calculate the mean value and standard deviation of these $\theta_k$ values over all samples on each interval. The results are depicted in Fig. 2(e-h).

## 3.3 OPTIMIZING THE SHAPING FUNCTION WITHOUT OOD SAMPLES

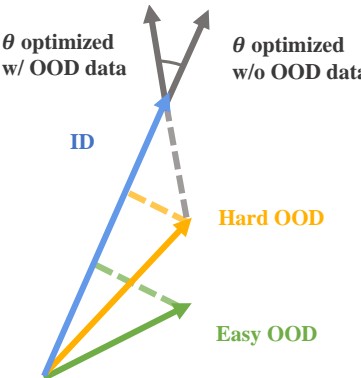

**θ optimized w/ OOD data**

**θ optimized w/o OOD data**

**ID**

**Hard OOD**

**Easy OOD**

Figure 3: Diagram to show the intuition in deriving Eq. 13.

Normally, we have access to the ID training set, enabling us to estimate $\mathbf{E}_{\boldsymbol{x}^{\text{ID}} \sim D_{\mathcal{X}}^{\text{ID}}}\left[\boldsymbol{I}(\boldsymbol{z}^{\text{ID}})\right]$. However, a significant challenge arises with the unknown term $\mathbf{E}_{\boldsymbol{x}^{\text{OOD}} \sim D_{\mathcal{X}}^{\text{OOD}}}\left[\boldsymbol{I}(\boldsymbol{z}^{\text{OOD}})\right]$, which renders the optimization problem defined in Eq. 10 intractable.

To make the problem tractable without access to OOD data, we omit the OOD-related term $\mathbf{E}_{\boldsymbol{x}^{\text{OOD}} \sim D_{\mathcal{X}}^{\text{OOD}}}\left[\boldsymbol{I}(\boldsymbol{z}^{\text{OOD}})\right]$ by the following argument, which as we will see is also supported empirically in Section 4.2.

A diagram to explain our intuition is shown in Fig. 3. Specifically, in scenarios where the OOD distribution closely resembles the ID (hard OOD), OOD samples likely have similar feature representations to ID samples, implying both expectations share similar directions. Conversely, in cases where OOD samples significantly diverge from ID samples (easy OOD), the maximum logits of OOD samples tend to be lower, thereby reducing the magnitude of $\mathbf{E}_{\boldsymbol{x}^{\text{OOD}} \sim D_{\mathcal{X}}^{\text{OOD}}}\left[\boldsymbol{I}(\boldsymbol{z}^{\text{OOD}})\right]$ compared to ID.

Thus, the component of $\mathbf{E}_{\boldsymbol{x}^{\text{OOD}} \sim D_{\mathcal{X}}^{\text{OOD}}}\left[\boldsymbol{I}(\boldsymbol{z}^{\text{OOD}})\right]$ orthogonal to ID expectation is small and we suggest omitting $\mathbf{E}_{\boldsymbol{x}^{\text{OOD}} \sim D_{\mathcal{X}}^{\text{OOD}}}\left[\boldsymbol{I}(\boldsymbol{z}^{\text{OOD}})\right]$, resulting in the following formulation:

$$
\begin{aligned}
\text{maximize (over } \boldsymbol{\theta}) \quad & \boldsymbol{\theta}^{\mathsf{T}} \mathbf{E}_{\boldsymbol{x} \sim D_{\mathcal{X}}^{\text{ID}}}[\boldsymbol{I}(\boldsymbol{z})] \\
\text{subject to} \quad & \|\boldsymbol{\theta}\| = \sqrt{K}, \quad \boldsymbol{z} = \mathcal{B}(x).
\end{aligned}
\tag{13}
$$

The optimal solution to this new problem can be expressed in closed-form as:

$$
\boldsymbol{\theta}^{\star} = \frac{\sqrt{K}}{\left\|\mathbf{E}_{\boldsymbol{x} \sim D_{\mathcal{X}}^{\text{ID}}}[\boldsymbol{I}(\boldsymbol{z})]\right\|_2} \mathbf{E}_{\boldsymbol{x} \sim D_{\mathcal{X}}^{\text{ID}}}[\boldsymbol{I}(\boldsymbol{z})].
\tag{14}
$$

We plot $\boldsymbol{\theta}^{\star}$ calculated by Eq. 14 in Fig. 2(d), noting that it is in close proximity to the optimal solution in the case where OOD samples are available. In experiments, we will also show that our method with only ID data performs almost on par with using the true $\boldsymbol{I}(\boldsymbol{z}^{\text{OOD}})$ calculated from OOD data.

At inference time, for each sample, we rescale its features within each interval $[a_k, b_k)$ by the corresponding entry in $\boldsymbol{\theta}^{\star}$, and calculate an OOD score based on the reshaped feature representation.

## 4 EXPERIMENTS

**Datasets.** We evaluate our proposed method using a large scale benchmark where ImageNet-1k (Russakovsky et al., 2015) is the ID dataset, as well as two moderate scale benchmarks where CIFAR10 and CIFAR100 (Krizhevsky et al., 2009) are used as the ID datasets. We evaluate eight OOD datasets for the ImageNet benchmark, comprised of Species (Hendrycks et al., 2019b), iNaturalist (Van Horn et al., 2018), SUN (Xiao et al., 2010), Places (Zhou et al., 2017), OpenImage-O (Wang et al., 2022), ImageNet-O (Hendrycks et al., 2021), Texture (Cimpoi et al., 2014), and MNIST (Deng, 2012). When CIFAR10/100 is the ID dataset, eight OOD datasets are exploited, comprised of TinyImageNet (Torralba et al., 2008), SVHN (Netzer et al., 2011), Texture (Cimpoi et al., 2014), Places365 (Zhou et al., 2017), LSUN-Cropped (Yu et al., 2015), LSUN-Resized (Yu et al., 2015), iSUN (Xu et al., 2015), and finally CIFAR100/10.

**Model architectures.** We use pre-trained models provided by PyTorch and prior works in our experiments. For ImageNet-1k, we use a pre-trained ResNet50 (He et al., 2016) provided by PyTorch and a MobileNet-v2 (Sandler et al., 2018) provided by Djurisic et al. (2022). Additionally, we include models based on the transformer and fully-MLP architectures, which have been largely overlooked in previous studies. Specifically, we include ViT-B-16, ViT-L-16 (Dosovitskiy et al., 2020), SWIN-S, SWIN-B (Liu et al., 2021), MLP-Mixer-B, and MLP-Mixer-L (Tolstikhin et al., 2021). For the CIFAR10 and CIFAR100 benchmarks, we use a DenseNet101 (Huang et al., 2017) model provided by Sun & Li (2022), a ViT-B-16 finetuned on CIFAR10/100 consistent with Fort et al. (2021), and a MLP-Mixer-Nano model trained on CIFAR10/100 from scratch.

Table 1: OOD detection results on ImageNet-1k. ↑ indicates larger values are better and ↓ indicates smaller values are better. All values are percentages averaged over different OOD datasets. **Bold** numbers are superior results whereas underlined numbers denote the second and third best results.

| Method | ResNet50 FP↓ | AU↑ | MobileNetV2 FP↓ | AU↑ | ViT-B-16 FP↓ | AU↑ | ViT-L-16 FP↓ | AU↑ | SWIN-S FP↓ | AU↑ | SWIN-B FP↓ | AU↑ | MLP-B FP↓ | AU↑ | MLP-L FP↓ | AU↑ | Average FP↓ | AU↑ |
|---|---|---|---|---|---|---|---|---|---|---|---|---|---|---|---|---|---|---|
| MSP | 69.30 | 76.26 | 72.58 | 77.41 | 69.84 | 77.40 | _70.59_ | 78.40 | 65.12 | 80.89 | 69.45 | 77.26 | _70.66_ | _80.31_ | _71.63_ | _79.93_ | 69.90 | 78.48 |
| ODIN[a] | 61.56 | 80.92 | 62.91 | 82.64 | **69.25** | 72.60 | _70.35_ | 74.51 | _62.35_ | 78.67 | 70.90 | 68.36 | _65.00_ | **82.99** | _67.90_ | **81.86** | _66.28_ | 77.82 |
| Energy | 60.97 | 81.01 | 61.40 | 82.83 | 73.96 | 67.65 | 74.89 | 70.11 | 66.56 | 75.95 | 79.53 | 60.14 | 87.86 | 78.42 | 84.60 | 79.01 | 73.72 | 74.39 |
| DICE | 45.32 | 83.64 | _49.33_ | 84.63 | 89.68 | 71.32 | 72.38 | 67.08 | 78.84 | 49.06 | 77.89 | 50.10 | **57.12** | _80.60_ | _61.98_ | _80.75_ | _66.57_ | 70.90 |
| ReAct | 42.29 | 86.54 | 54.19 | 85.37 | 73.82 | 76.86 | 76.16 | 81.07 | **58.07** | _85.10_ | 70.23 | 80.10 | 90.13 | 77.18 | 89.03 | 78.32 | 69.24 | _81.32_ |
| BFAct | 43.87 | 86.01 | 52.87 | 85.78 | 77.64 | _80.16_ | 84.02 | _81.12_ | 85.57 | 68.40 | 84.36 | 69.04 | 96.04 | 67.87 | 96.00 | 70.59 | 72.95 | 80.18 |
| ASH-P | 55.30 | 83.00 | 59.41 | 83.84 | 99.36 | 21.17 | 99.18 | 20.27 | 99.09 | 21.78 | 98.83 | 22.11 | 98.19 | 35.85 | 98.91 | 29.69 | 88.53 | 39.71 |
| ASH-B | _35.97_ | _88.62_ | _43.59_ | **88.28** | 94.87 | 46.68 | 93.72 | 38.95 | 96.48 | 36.54 | 92.40 | 49.85 | 99.51 | 21.73 | 64.91 | 83.15 | 77.68 | 56.73 |
| ASH-S | **34.70** | **90.25** | 43.84 | _88.24_ | 99.48 | 18.52 | 99.42 | 18.61 | 99.20 | 18.26 | 99.06 | 19.27 | 97.61 | 33.91 | 99.28 | 19.29 | 84.07 | 38.29 |
| VRA-P | _37.97_ | _88.58_ | 49.98 | _86.83_ | 98.39 | 35.66 | 99.58 | 16.70 | 99.27 | 20.34 | 99.46 | 17.64 | 99.38 | 18.73 | 99.05 | 21.23 | 85.39 | 38.21 |
| Ours (V) | 41.84 | 88.11 | 53.31 | 86.36 | _69.33_ | 82.59 | _72.17_ | **83.23** | 66.89 | 84.53 | **65.96** | _83.71_ | 76.59 | 78.46 | 78.13 | 78.85 | **65.53** | **83.23** |
| Ours (E) | 39.75 | 88.56 | 51.77 | 86.62 | _69.52_ | **82.66** | 78.57 | 82.94 | 66.28 | 84.99 | _66.20_ | _84.21_ | 83.54 | 75.94 | 85.50 | 78.11 | _67.64_ | _83.00_ |

[a] Choice of hyperparameters on Imagenet makes ODIN yield identical results to MLS.

Table 2: OOD detection results on CIFAR10/100. ↑ indicates larger values are better and ↓ indicates smaller values are better. All values are percentages averaged over different OOD datasets. **Bold** numbers are superior results whereas underlined numbers denote the second and third best results.

| Method | CIFAR10 DenseNet101 FP↓ | AU↑ | ViT-B-16 FP↓ | AU↑ | MLP-N FP↓ | AU↑ | Average FP↓ | AU↑ | CIFAR100 DenseNet101 FP↓ | AU↑ | ViT-B-16 FP↓ | AU↑ | MLP-N FP↓ | AU↑ | Average FP↓ | AU↑ |
|---|---|---|---|---|---|---|---|---|---|---|---|---|---|---|---|---|
| MSP | 52.66 | 91.42 | 25.47 | 95.61 | 67.01 | _82.86_ | 48.38 | _89.96_ | 80.40 | 74.75 | 61.24 | 85.24 | 83.97 | 73.07 | 75.20 | 77.69 |
| ODIN | 32.84 | 91.94 | 24.93 | 92.84 | 69.20 | 69.53 | 42.32 | 84.77 | 62.03 | 82.57 | 56.91 | 80.86 | **78.71** | 66.47 | **65.88** | 76.63 |
| MLS | 31.93 | 93.51 | 12.79 | 97.16 | 64.50 | **82.97** | _36.41_ | **91.21** | 70.71 | 80.18 | 52.31 | 88.63 | 82.41 | 74.52 | 68.48 | 81.11 |
| Energy | 31.72 | 93.51 | _12.40_ | _97.22_ | **63.95** | _82.84_ | **36.02** | _91.19_ | 70.80 | 80.22 | _51.93_ | 88.82 | _79.90_ | _75.30_ | 67.54 | _81.45_ |
| DICE | 29.67 | 93.27 | 26.84 | 92.41 | 96.64 | 52.17 | 51.05 | 79.28 | _59.56_ | 82.26 | 95.18 | 42.05 | 95.78 | 44.48 | 83.51 | 56.26 |
| ReAct | 82.00 | 76.46 | **12.33** | **97.29** | _64.34_ | 81.85 | 52.89 | 85.20 | 77.00 | 78.30 | 52.25 | _88.84_ | 79.99 | _75.87_ | 69.75 | 81.00 |
| BFAct | 84.40 | 74.39 | _12.34_ | _97.21_ | 78.02 | 72.68 | 58.25 | 81.43 | 80.27 | 73.36 | **51.23** | **89.04** | 80.05 | **76.58** | 70.52 | 79.66 |
| ASH-P | 29.39 | 93.98 | 20.20 | 95.31 | 84.39 | 66.93 | 44.66 | 85.41 | 68.21 | 81.11 | 53.69 | 87.66 | 86.73 | 65.27 | 69.54 | 78.01 |
| ASH-B | _28.21_ | _94.27_ | 82.10 | 69.54 | 93.93 | 53.00 | 68.08 | 72.27 | _57.45_ | _83.80_ | 91.74 | 56.39 | 93.63 | 57.20 | 80.94 | 65.80 |
| ASH-S | **23.93** | **94.41** | 39.93 | 91.06 | 82.57 | 68.02 | 48.81 | 84.50 | 52.41 | **84.65** | — | 84.25 | 89.39 | 59.63 | _67.41_ | 76.18 |
| VRA-P | 38.41 | 92.77 | 13.73 | 96.86 | 100.00 | 65.95 | 50.71 | 85.19 | 67.75 | _82.72_ | 52.21 | 88.49 | 87.19 | 66.03 | 69.05 | 79.08 |
| Ours (V) | 30.12 | 93.97 | 13.13 | 97.07 | 77.25 | 73.86 | _40.17_ | 88.30 | 69.01 | 81.42 | 52.76 | 88.60 | 83.83 | 73.65 | 68.53 | _81.22_ |
| Ours (E) | _28.90_ | _94.12_ | 12.79 | 97.14 | 83.87 | 71.83 | _41.85_ | 87.70 | 65.2 | 82.39 | _51.67_ | 88.78 | 81.33 | 74.67 | _66.07_ | **81.95** |

**Evaluation metrics.** We utilize two standard evaluation metrics, following previous works (Liang et al., 2017; Sun et al., 2021; Sun & Li, 2022): the false positive rate (FPR95) when the true positive rate of the ID samples is 95%, and AUC.

**Implementation details.** We evaluate two variants of our method, the vanilla method described in Eq. 14 denoted "Ours (V)", based on the maximum logit score, and a variant based on energy scores, denoted "Ours (E)". For comparison, we implement five baseline methods (top rows in results tables), including MSP (Hendrycks & Gimpel, 2016), ODIN (Liang et al., 2017), MLS (Hendrycks et al., 2019a), Energy (Liu et al., 2020), and DICE (Sun & Li, 2022), and six feature-shaping methods (remaining rows in results tables), including ReAct (Sun et al., 2021), BFAct (Kong & Li, 2023), ASH-P, ASH-B, ASH-S (Djurisic et al., 2022), and VRA-P (Xu & Lian, 2023). All comparison feature-shaping methods are evaluated using the energy score. Hyperparameters are set consistent with the original papers and concrete hyperparameter settings can be found in Section A.1 of the Appendix. All experiments are run on a single NVIDIA GeForce RTX 4090 GPU.

## 4.1 OOD DETECTION BENCHMARK RESULTS

Experimental results on the ImageNet-1k benchmark are shown in Table 1. More detailed results can be found in Sections E and F of the Appendix. We observe that the comparison methods for

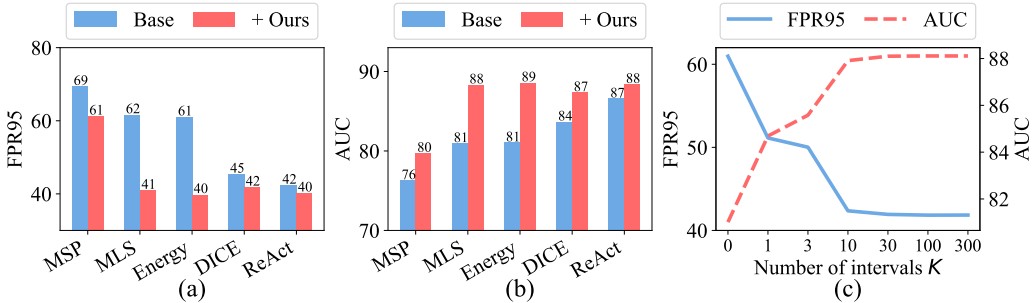

Figure 4: Compatibility and sensitivity analysis. (a-b) Our method can improve other OOD scores and methods. "Base" denotes using the original OOD score or method, while "+Our" indicates combining the score or method with our feature-shaping function. (c) Our method's performance with different hyperparameter settings, i.e., numbers of intervals $K$.

feature shaping perform extremely well on the pretrained ResNet-50 model, with some ASH variants outperforming element-wise shaping methods like ReAct (Sun et al., 2021), and yielding similar performance to our proposed methods. This is not surprising, given that ResNet-50 was the model architecture used in the original papers. As we showed in Fig. 2, the shaping functions proposed by these methods closely align with the optimal shaping function learned given actual OOD samples.

However, we observe that these methods do not generalize well to architectures outside of ResNet-50. We observe that for feature-shaping methods more generally, stronger convolutional network architecture performance is negatively correlated to performance on other network architectures. However, our proposed methods yield consistently strong performance across all network architectures, despite only having access to ID data for parameter tuning.

Results for the CIFAR10 and CIFAR100 benchmarks are shown in Table 2. An important observation regarding these results is that feature-shaping methods as a whole seem to underperform the baseline methods. This indicates that feature shaping appears to be ineffective on smaller-scale datasets such as CIFAR. While our methods do not outperform the baselines, we are competitive and do outperform all of the comparison feature shaping methods, which is consistent with the ImageNet-1k benchmark.

## 4.2 FURTHER ANALYSIS

We conduct additional experiments using the ImageNet-1k benchmark and employing ResNet50.

**Extension to other OOD scores.** While our proposed method in Eq. 14 is designed assuming MLS as the OOD score, we also extend our method to use other OOD scores such as energy and MSP. Furthermore, we combine our method with other OOD detection methods including DICE and ReAct. Results are averaged across the eight OOD datasets. As shown in Fig. 4(a-b), our method yields consistent improvement across these OOD detection scores and methods.

**Different numbers of intervals.** We evaluate our methods with different number of intervals, i.e., varying $K$. As shown in Fig. 4(c), most of the improvement in performance comes from $K \leq 10$. This indicates that the optimal feature shaping function has a relatively simple form. This is consistent with prior methods (Sun et al., 2021; Xu & Lian, 2023; Kong & Li, 2023), which assume 2–3 intervals. In addition, the relatively large performance gap over the baseline motivates feature shaping more generally. Interestingly, we observe an extremely large jump between $K = 0$ ("Energy" baseline) with $K = 1$, which effectively prunes the top and bottom 0.1% of feature values (using the thresholds computed based on the ID training data).

**Empirical validation of Problem 13.** In deriving Eq. 13, we remove the OOD-related term, assuming OOD data is inaccessible. We now empirically justify our intuition shown in Fig. 3. We estimate $\boldsymbol{a} = \mathbf{E}_{\boldsymbol{x}^{\mathrm{ID}} \sim D_{\mathcal{X}}^{\mathrm{ID}}}\left[\boldsymbol{I}(\boldsymbol{z}^{\mathrm{ID}})\right]$ with ImageNet-1k and $\boldsymbol{b} = \mathbf{E}_{\boldsymbol{x}^{\mathrm{OOD}} \sim D_{\mathcal{X}}^{\mathrm{OOD}}}\left[\boldsymbol{I}(\boldsymbol{z}^{\mathrm{OOD}})\right]$ with different OOD datasets. We then compute the angles and the norm ratios, $\frac{\|\boldsymbol{b}\|}{\|\boldsymbol{a}\|}$, of the two expectations.

As shown in Table 3, challenging OOD datasets like Species and ImageNet-O exhibit smaller angles between the two expectations, while simpler OOD datasets like iNaturalist and MNIST demonstrate smaller norm ratios. Thus, Eq. 13 provides a reasonable approximation to $\boldsymbol{\theta}$ solved by Eq. 10 with access to OOD data.

**Compare the formulation utilizing OOD data (Eq. 10) and our method (Eq. 14).** We optimize $\boldsymbol{\theta}$ by solving Eq. 10 with each OOD dataset and get the OOD detection performance on that dataset. The last two columns of Table 3 compare the performance of $\boldsymbol{\theta}$ solved from Eq. 10

Table 3: Empirical validation of Problem 13. $\boldsymbol{a}$ and $\boldsymbol{b}$ denote the expectations of $\boldsymbol{I}(\boldsymbol{z})$ for ID and OOD, respectively. "AU (Real)" refers to the performance of solving Eq. 10 with OOD data.

| OOD dataset | $\|\|\boldsymbol{b}\|\|/\|\|\boldsymbol{a}\|\|$ | $\cos(\boldsymbol{a}, \boldsymbol{b})$ | AU (Real) | AU (Ours) |
|---|---|---|---|---|
| Species | 0.72 | 0.99 | 78.69 | 78.59 |
| iNaturalist | 0.48 | 0.97 | 96.53 | 96.63 |
| SUN | 0.55 | 0.98 | 92.90 | 92.84 |
| Places | 0.59 | 0.99 | 90.06 | 90.15 |
| OpenImage-O | 0.56 | 0.98 | 92.31 | 92.54 |
| ImageNet-O | 0.91 | 0.99 | 70.19 | 59.33 |
| Texture | 0.51 | 0.96 | 95.82 | 95.48 |
| MNIST | 0.42 | 0.84 | 99.32 | 99.34 |

and that from Eq. 14 (ours). Notably, our method with only ID data yields performance comparable to the formulation with access to real OOD data.

## 5 DISCUSSION

Here we first discuss a core limitation to our framework that our proposed method assumes element-wise shaping functions, which fails to capture methods like ASH (Dosovitskiy et al., 2020). We then connect our proposed method to existing works.

**General feature-shaping functions.** As discussed previously, a feature-shaping function can represent arbitrary mappings $\boldsymbol{f} : \mathbb{R}^M \to \mathbb{R}^M$. While our proposed methods handle mappings which can be defined element-wise over individual features, they fail to capture methods such as ASH (Dosovitskiy et al., 2020) which can only be defined over the full feature vector. An important direction for future work is in designing a tractable optimization problem where the family of functions $\boldsymbol{f}$ includes whole-vector methods such as ASH. We expect that the resultant mappings should also be relatively simple as in the element-wise case.

**Feature contributions to the maximum logits.** Prior research (Sun & Li, 2022; Dietterich & Guyer, 2022) has explored the idea of feature contributions to the maximum logits. For example, DICE (Sun & Li, 2022) identifies the most significant feature indices for each class using ID data. During test-time, it masks feature indices deemed insignificant before calculating OOD scores. However, our approach differs from theirs in that we consider feature value ranges instead of indices. We find our function yields significantly better performance in practice, partially attributed to the fact that it admits a tractable optimization problem. Investigating the combination of these two classes of methods remains a promising direction for future work.

**Classifier training for OOD detection.** Prior studies (Liang et al., 2017; Sun & Li, 2022) have leveraged surrogate OOD datasets, such as Gaussian and Uniform distributions, to train classifiers for distinguishing between ID and OOD samples. Liang et al. (2017) primarily employ these datasets to assess their method's performance, while Sun & Li (2022) use them for hyperparameter tuning. Despite these efforts, the approach is seldom applied in feature-shaping methods. Concurrently, several studies are probing into outlier synthesis (Du et al., 2022; He et al., 2022) to approximate real OOD distributions and train classifiers accordingly. Our optimization problem could benefit from these advancements, as our parameterized feature-shaping function could be optimized based on an ID dataset and synthesized outliers. We explore this possibility in Section B.1 of the Appendix.

## 6 CONCLUSION

This paper formulates a general optimization problem for feature-shaping methods in OOD detection. Using this formalism, we are able to better understand and explain the effectiveness of existing state-of-the-art methods. In the context of our framework, we propose a novel feature shaping method which can be trained solely on ID data. Experimental results validate the superior performance and generalization ability of our approach compared to prior works, and motivate the development of new methods under our framework to further advance OOD detection methods.

ACKNOWLEDGMENTS

We would like to extend our deepest appreciation to Weijian Deng, Yanbin Liu, Yunzhong Hou, Xiaoxiao Sun, and all other lab colleagues for their invaluable support throughout this project. Their collaborative efforts, insightful discussion, and constructive feedback have been crucial in shaping and improving our paper.

This work was supported by an Australian Research Council (ARC) Linkage grant (project number LP210200931).

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

## A  IMPLEMENTATION DETAILS

### A.1  HYPERPARAMETER SETTINGS FOR EXPERIMENTS IN SECTION 4

For feature-shaping methods, we adhere to the hyperparameter settings reported in the original papers of the respective methods. These settings usually optimize the performance of the methods on existing benchmarks. For additional methods, we primarily rely on hyperparameter settings detailed in Sun's work (Sun & Li, 2022). The detailed hyperparameter setting are listed in Table 4.

Table 4: Hyperparameter settings for experiments in Section 4

| Method | Hyperparameters | ImageNet-1k | CIFAR10 | CIFAR100 |
|---|---|---|---|---|
| MSP | No hyperparameters | - | - | - |
| ODIN | $T$, temperature scaling $\epsilon$, noise level | $T = 1000, \epsilon = 0.0$ | $T = 1000, \epsilon = 0.004$ | $T = 1000, \epsilon = 0.004$ |
| MLS | No hyperparameters | - | - | - |
| Energy | No hyperparameters | - | - | - |
| DICE | $p$, sparsity parameter | $p = 0.7$ | $p = 0.9$ | $p = 0.9$ |
| ReAct | $p$, threshold percentile | $p = 90$ | $p = 90$ | $p = 90$ |
| BFAct | $thresh, N$ | $N = 2, thresh = 95$ percentile of the training set | $N = 2, thresh = 95$ percentile of the training set | $N = 2, thresh = 95$ percentile of the training set |
| ASH-P | $p$, pruning percentile | $p = 60$ | $p = 90$ | $p = 80$ |
| ASH-B | $p$, pruning percentile | $p = 65$ | $p = 95$ | $p = 85$ |
| ASH-S | $p$, pruning percentile | $p = 90$ | $p = 95$ | $p = 90$ |
| VRA-P | $x_1, x_2$ | $x_1 = 0.5, x_2 = 1.0$ | $x_1 = 60$ percentile of the training set $x_2 = 95$ percentile of the training set | $x_1 = 60$ percentile of the training set $x_2 = 95$ percentile of the training set |
| Ours | $K$, the number of feature value intervals | $K = 100$ | $K = 100$ | $K = 100$ |

### A.2  THE MAGNITUDE OF RESHAPED FEATURES

In Section 3, we describe our method for learning a piecewise constant reshaping function subject to norm constraints. Scaling $\boldsymbol{\theta}$ has no effect on the vanilla version of method. However, OOD detection performance is sensitive to the scaling of $\boldsymbol{\theta}$ when our method is combined with other OOD scores with non-linear transformations such as MSP and the energy score.

Take the energy score as an example. If we multiply the feature representation $\boldsymbol{z}$ by a small positive factor $q$ ($0 < q < 1$), ignoring the bias term in the last fully connected layer, we have:

$$s_{\text{Energy}}(\boldsymbol{\ell}) = \log \sum_{j=1}^{C} \exp(q\ell_j). \tag{15}$$

This is equivalent to apply a temperature scaling $T = 1/q$ to the energy score, which will affect the OOD detection performance, as analyzed in a previous paper (Liang et al., 2017). Besides, considering a fully connected layer usually contains a bias term, the scaling of $\boldsymbol{\theta}$ will further affect the influence of the bias. Thus, we need to consider the scale of $\boldsymbol{\theta}$ in our optimization problem.

We scale $\boldsymbol{\theta}$ by a factor of $\sqrt{K}$, motivated by the following analysis. Considering that the maximum logit is the sum of all ISFIs, we have:

$$\ell^{\max}(\boldsymbol{z}) = \sum_{k=1}^{K} I_{[a_k, b_k]}(\boldsymbol{z}) = \mathbf{1}^{\top} \boldsymbol{I}(\boldsymbol{z}), \tag{16}$$

where $\mathbf{1}$ represents a $K$-dimensional vector of ones, which is a trivial feature-shaping function with no effect. Therefore, we set $\|\boldsymbol{\theta}\| = \|\mathbf{1}\| = \sqrt{K}$.

Table 5: OOD detection results of variants of our methods on ImageNet-1k. ↑ indicates larger values are better and ↓ indicates smaller values are better. All values are percentages averaged over different OOD datasets. **Bold** numbers are superior results whereas underlined numbers denote the second and third best results. "Real" means using a real OOD dataset to resolve Problem (Eq. 10). "Gaussian" and "Uniform" indicate using Gaussian and Uniform distributions as surrogate OOD datasets in resolving Problem (Eq. 10), respectively. "Dynamic" denotes solving a non-convex problem with dynamic weight indices defined in Section B.2.

| Method | ResNet50 FP↓ | AU↑ | MobileNetV2 FP↓ | AU↑ | ViT-B-16 FP↓ | AU↑ | ViT-L-16 FP↓ | AU↑ | SWIN-S FP↓ | AU↑ | SWIN-B FP↓ | AU↑ | MLP-B FP↓ | AU↑ | MLP-L FP↓ | AU↑ | Average FP↓ | AU↑ |
|---|---|---|---|---|---|---|---|---|---|---|---|---|---|---|---|---|---|---|
| MSP | 69.30 | 76.26 | 72.58 | 77.41 | 69.84 | 77.40 | 70.59 | 78.40 | 65.12 | 80.89 | 69.45 | 77.26 | 70.66 | 80.31 | 71.63 | 79.93 | 69.90 | 78.48 |
| ODIN | 61.56 | 80.92 | 62.91 | 82.64 | **69.25** | 72.60 | **70.35** | 74.51 | 62.35 | 78.67 | 70.90 | 68.36 | 65.00 | **82.99** | 67.90 | 81.86 | 66.28 | 77.82 |
| Energy | 60.97 | 81.01 | 61.40 | 82.83 | 73.96 | 67.65 | 74.89 | 70.11 | 66.56 | 75.95 | 79.53 | 60.14 | 87.86 | 78.42 | 84.60 | 79.01 | 73.72 | 74.39 |
| DICE | 45.32 | 83.64 | 49.33 | 84.63 | 89.68 | 71.32 | 72.38 | 67.08 | 78.84 | 49.06 | 77.89 | 50.10 | **57.12** | 80.60 | **61.98** | 80.75 | 66.57 | 70.90 |
| ReAct | 42.29 | 86.54 | 54.19 | 85.37 | 73.82 | 76.86 | 76.16 | 81.07 | **58.07** | 85.10 | 70.23 | 80.10 | 90.13 | 77.18 | 89.03 | 78.32 | 69.24 | 81.32 |
| BFAct | 43.87 | 86.01 | 52.87 | 85.78 | 77.64 | 80.16 | 84.02 | 81.12 | 64.76 | **85.57** | 68.40 | **84.36** | 96.04 | 67.87 | 96.00 | 70.59 | 72.95 | 80.18 |
| ASH-P | 55.30 | 83.00 | 59.41 | 83.84 | 99.36 | 21.17 | 99.18 | 20.27 | 99.09 | 21.78 | 98.83 | 22.11 | 98.19 | 35.85 | 98.91 | 29.69 | 88.53 | 39.71 |
| ASH-B | 35.97 | 88.62 | **43.59** | **88.28** | 94.87 | 46.68 | 93.72 | 38.95 | 96.48 | 36.54 | 92.40 | 49.85 | 99.51 | 21.73 | 64.91 | **83.15** | 77.68 | 56.73 |
| ASH-S | **34.70** | **90.25** | 43.84 | 88.24 | 99.48 | 18.52 | 99.42 | 18.61 | 99.20 | 18.26 | 99.06 | 19.27 | 97.61 | 33.91 | 99.28 | 19.29 | 84.07 | 38.29 |
| VRA-P | 37.97 | 88.58 | 49.98 | 86.83 | 98.39 | 35.66 | 99.58 | 16.70 | 99.27 | 20.34 | 99.46 | 17.64 | 99.38 | 18.73 | 99.05 | 21.23 | 85.39 | 38.21 |
| Real | 42.09 | 88.13 | 55.05 | 85.77 | 73.69 | 81.65 | 75.36 | 83.10 | 76.06 | 83.22 | 73.54 | 83.11 | 81.66 | 78.34 | 78.82 | 79.20 | 69.53 | 82.82 |
| Gaussian | 42.05 | 88.20 | 55.35 | 85.67 | 71.88 | 82.01 | 73.99 | **83.28** | 71.29 | 83.99 | 70.79 | 83.37 | 82.09 | 77.58 | 76.40 | 79.79 | 67.98 | 82.99 |
| Uniform | 42.36 | 88.02 | 55.12 | 85.73 | 73.14 | 81.82 | 74.30 | 83.25 | 72.68 | 83.74 | 70.57 | 83.43 | 80.84 | 76.88 | 76.41 | 79.74 | 68.18 | 82.83 |
| Dynamic | 41.81 | 88.13 | 53.29 | 86.35 | 69.33 | 82.59 | 72.18 | 83.23 | 66.90 | 84.52 | 65.98 | 83.71 | 76.87 | 78.35 | 78.10 | 78.85 | 65.56 | 83.22 |
| Ours (V) | 41.84 | 88.11 | 53.31 | 86.36 | 69.33 | 82.59 | 72.17 | 83.23 | 65.96 | 83.71 | **65.53** | **83.23** |
| Ours (E) | 39.75 | 88.56 | 51.77 | 86.62 | 69.52 | **82.66** | 78.57 | 82.94 | 66.28 | 84.99 | 66.20 | 84.21 | 83.54 | 75.94 | 85.50 | 78.11 | 67.64 | 83.00 |

# B  VARIANTS OF OUR METHOD

In this section, we execute a series of experiments on variants of our proposed methods. Initially, we employ real and surrogate OOD datasets to address the optimization problem (Eq. 10) as alternatives to our ID only method. Subsequently, we evaluate an assumption that weight indices of maximum logit will not change after feature shaping. We calculate and report the ratio of changed weights after reshaping. Furthermore, we solve an optimization problem which allows for dynamic weight indices. In this scenario, it is possible that the weight indices of the maximum logits can undergo changes after the reshaping exercise. Our experiments are based on the ImageNet-1k benchmark and the results are illustrated in Table 5.

## B.1  EXPLOIT REAL AND SURROGATE OOD DATASETS

We draw comparisons between our ID only method and a feature-shaping function optimization approach that accesses both ID and OOD datasets.

First, we utilize real OOD datasets to address Problem (Eq. 10). For each model architecture, ImageNet-1k training set (ID) and iNaturalist (OOD) are leveraged to optimze $\theta$.

Second, we incorporate Gaussian and Uniform distributions as surrogate OOD datasets in resolving Problem (Eq. 10), aligning with prior studies (Liang et al., 2017; Sun & Li, 2022). For the Gaussian distribution, each pixel is generated with a mean of 127.5 and a standard deviation of 255. For the uniform distribution, we directly generate random pixel values within the range [0, 255]. Every generated image is of size $32 \times 32$, and is subsequently clipped and rounded to conform to the [0, 255] value range. For each surrogate OOD dataset, we generate a total of 50,000 samples.

As illustrated in Table 5, our method, even without access to OOD data, performs comparably to methods that optimize $\theta$ utilizing both ID and OOD datasets. Moreover, we observe that a more robust objective function, such as Log Loss or Hinge Loss, could potentially outperform our method when given access to real or surrogate OOD datasets. This presents an intriguing avenue for exploration.

Table 6: Ratio of changed weight indices $r$ on ImageNet-1k. All numbers are percentages. We randomly sample 10,000 samples from ImageNet-1k training set, and $r$ is defined in Section B.2

|   | ResNet50 | MobileNetV2 | ViT-B-16 | ViT-L-16 | SWIN-S | SWIN-B | MLP-B | MLP-L | Average |
|---|----------|-------------|----------|----------|--------|--------|-------|-------|---------|
| r | 6.8 | 6.5 | 1.1 | 0.5 | 1.5 | 0.9 | 4.3 | 0.4 | 2.8 |

## B.2 DYNAMIC WEIGHT INDICES

Our proposed method, as elucidated in Sections 3.2 and 3.3, relies on a fundamental simplifying assumption: the class index of the maximum logit remains unchanged after feature shaping.

We initially examine the validity of this assumption by calculating the ratio of weight indices that change after feature shaping. For a given ID training set $S_{\text{train}}$ of $N$ samples, each sample $\boldsymbol{x}^{(i)} \in S_{\text{train}}$ has weight indices corresponding to the maximum logits before and after feature shaping, as follows:

$$\tilde{k}^{(i)} = \arg\max_k \boldsymbol{w}_k^\mathsf{T} \boldsymbol{z}^{(i)} \quad \bar{k}^{(i)} = \arg\max_k \boldsymbol{w}_k^\mathsf{T} \bar{\boldsymbol{z}}^{(i)}, \tag{17}$$

where $w_k$ denotes the weight vector corresponding to class $k$ in the final fully-connected layer, $\boldsymbol{z}^{(i)}$ represents the feature representation for the sample $\boldsymbol{x}^{(i)}$, and $\bar{\boldsymbol{z}}^{(i)}$ is the reshaped feature. Subsequently, we compute the ratio of changed weight indices as:

$$r = \frac{\sum_{i=1}^N \mathbb{1}\left\{\tilde{k}^{(i)} \neq \bar{k}^{(i)}\right\}}{N} \times 100\%. \tag{18}$$

We randomly sample 10,000 samples from ImageNet-1k training set. The ratios of changed weights with different models are presented in Table 6. As observed, the ratios are markedly small, which further substantiates the credibility of our assumption pertaining to fixed weights.

Furthermore, we propose a straightforward Alternating Optimization algorithm for dynamic weight indices. Each iteration involves three primary steps. Firstly, we optimize our feature-shaping function by resolving the optimization problem (Eq. 10) given the weights relative to the maximum logits for each sample. Next, we reshape features utilizing our optimized feature-shaping function. Lastly, we update the weights with reshaped features. To expedite the process, we randomly subsample 10,000 samples from the ID training set during each iteration and utilize them for optimization. Ten iterations are run for each model architecture. The results can be viewed in Table 5.

Predictably, this approach yields performance metrics that closely mirror our initial method. Despite the incorporation of dynamic weight indices, the optimized feature-shaping function does not significantly differ from that obtained with fixed weights. This observation reinforces the validity of our original assumption: the class index of the maximum logit remains constant following reshaping.

## C ROBUSTNESS OF OUR METHOD TO THE CHOSEN LOWER AND UPPER LIMITS OF FEATURES

Given that feature distributions typically exhibit long-tailed characteristics, we select the 0.1 and 99.9 percentiles of all features in a training set as the lower and upper limits of feature values, respectively, to optimize our shaping function. Here we validate different choices of percentiles on the performance of our method, on the ImageNet-1k benchmark with ResNet50. The results are shown in Table 7.

Our method exhibits a insensitivity to the chooses of lower and upper limits. Although in the main experiments we consistently use the 0.1 and 99.9 percentiles, exploring alternate settings reveals potential for slight improvements.

Table 7: Performance of our method with different choices of percentiles of all features on the ImageNet-1k benchmark with ResNet50.

| Lower percentile | Lower limit | Upper percentile | Upper limit | FPR95 | AUC |
|------------------|-------------|------------------|-------------|-------|-----|
| 0.0 | 0.00 | 100.0 | 20.65 | 42.42 | 87.90 |
| 0.01 | 0.00 | 99.99 | 5.64 | 42.22 | 87.97 |
| 0.05 | 0.00 | 99.95 | 4.40 | 41.97 | 88.04 |
| 0.1 | 0.00 | 99.9 | 3.88 | 41.82 | 88.11 |
| 0.5 | 0.00 | 99.5 | 2.74 | 41.01 | 88.44 |
| 1.0 | 0.00 | 99.0 | 2.28 | **41.00** | **88.50** |
| 5.0 | 0.03 | 95.0 | 1.35 | 47.85 | 86.41 |
| 10.0 | 0.06 | 90.0 | 1.01 | 57.45 | 83.60 |

## D    EMPIRICAL ANALYSIS OF THE OPTIMAL SHAPING FUNCTION

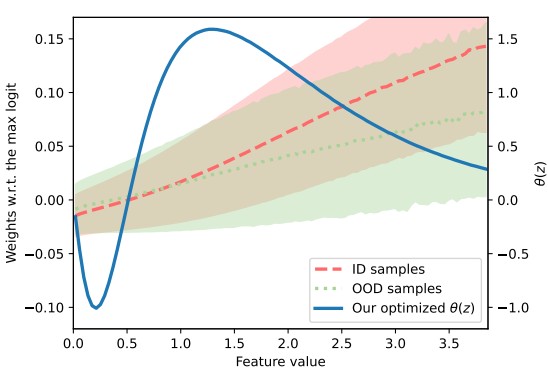

Here we try to provide empirical insights into the specific form of our optimal shaping function based on a ResNet50 model pretrained on ImageNet-1k, which is shown in Fig. 2. Specifically, we feed ImageNet-1k val (ID) and iNaturalist (OOD) samples to the model, getting the feature representation $z \in \mathbb{R}^{M}$ and the weights w.r.t. the maximum logit $w^{\max} \in \mathbb{R}^{M}$ for each sample. The results are shown in Fig. 5. The red and green lines represent the mean weights assigned to different feature values for ID and OOD samples, respectively. The colored regions represent standard deviations.

Figure 5: Empirical analysis to explain a specific form of the optimal shaping function.

An interesting aspect is that our approach flips the sign of low-value features, contrary to previous techniques like ReAct and ASH. Our experiments revealed that low-value features ($z < 0.5$) often align with negative weights ($w^{\max} < 0$). Intriguingly, ID samples generally align these features with smaller weights than OOD samples (on average, $w_{\text{ID}}^{\max} < w_{\text{OOD}}^{\max} < 0$). By flipping the sign of these low-value features, we can enhance the maximum logits more significantly for ID samples than for OOD samples, thereby boosting OOD detection performance.

However, note that the shape of the optimal shaping function is influenced by multiple factors. For instance, while ID samples are generally assigned larger weights for high-value features, OOD samples tend to exhibit a greater number of these features, as observed in prior studies (Sun et al., 2021). This aspect reduces the impact of high-value features in distinguishing between ID and OOD samples, leading to the decrease of $\theta$ for those features.

Besides, note that our analysis above is grounded in the context of a ResNet50 model pre-trained on ImageNet-1k. The form of the optimal shaping function may differ when applied to other model architectures or datasets.

## E    DETAILED OOD DETECTION PERFORMANCE

We report additional results in Tables 8, 9, and 10 to supplement the results provided in Table 1. Specifically, we provide detailed performance for all eight test OOD datasets for the ImageNet-1k benchmark for three representative models, including ResNet50, ViT-B-16, and MLP-B.

## F    STANDARD IMAGENET-1K BENCHMARK COMPARISON

Standard ImageNet-1k benchmark uses a subset of OOD datasets of ours, and these results can be found in Tables 8, 9, and 10.

Besides, we pick the results of standard ImageNet-1k benchmark in our experiments and calculated the new average performance. Our method still outperforms prior works on this standard benchmark. Results are shown in Table 11.

Table 8: OOD detection results with ResNet50 on ImageNet-1k. ↑ indicates larger values are better and ↓ indicates smaller values are better. All values are percentages averaged over different OOD datasets. **Bold** numbers are superior results whereas underlined numbers denote the second and third best results. "Real" means using a real OOD dataset to resolve Problem (Eq. 10). "Gaussian" and "Uniform" indicate using Gaussian and Uniform distributions as surrogate OOD datasets in resolving Problem (Eq. 10), respectively. "Dynamic" denotes solving a non-convex problem with dynamic weight indices defined in Section B.2.

| Method | Species | | iNaturalist | | SUN | | Places | | OpenImage-O | | ImageNet-O | | Texture | | MNIST | | Average | |
|---|---|---|---|---|---|---|---|---|---|---|---|---|---|---|---|---|---|---|
| | FP↓ | AU↑ | FP↓ | AU↑ | FP↓ | AU↑ | FP↓ | AU↑ | FP↓ | AU↑ | FP↓ | AU↑ | FP↓ | AU↑ | FP↓ | AU↑ | FP↓ | AU↑ |
| MSP | 79.53 | 75.15 | 52.77 | 88.42 | 68.58 | 81.75 | 71.57 | 80.63 | 63.58 | 84.98 | 100.00 | 28.64 | 66.13 | 80.46 | 52.25 | 90.02 | 69.30 | 76.26 |
| ODIN | 80.90 | 72.88 | 50.86 | 91.14 | 59.89 | 86.59 | 65.67 | 84.18 | 57.29 | 89.23 | 100.00 | 40.86 | 54.31 | 86.40 | 23.54 | 96.11 | 61.56 | 80.92 |
| Energy | 82.40 | 72.03 | 53.93 | 90.59 | 58.27 | 86.73 | 65.40 | 84.13 | 57.21 | 89.12 | 100.00 | 41.92 | 52.29 | 86.73 | 18.23 | 96.79 | 60.97 | 81.01 |
| DICE | 74.64 | 74.30 | 26.66 | 94.49 | 36.09 | 90.98 | 47.66 | 87.73 | 45.13 | 88.77 | 98.10 | 43.01 | 32.46 | 90.46 | 1.79 | 99.40 | 45.32 | 83.64 |
| ReAct | 68.64 | 77.41 | 19.55 | 96.39 | 24.00 | 94.41 | 33.43 | 91.93 | 43.68 | 90.53 | 98.05 | 52.43 | 45.83 | 90.45 | 5.11 | 98.76 | 42.29 | 86.54 |
| BFAct | 68.08 | 77.76 | 20.53 | 96.22 | 20.94 | 95.24 | 30.17 | 92.62 | 49.71 | 86.48 | 96.20 | 54.15 | 54.01 | 87.84 | 11.33 | 97.78 | 43.87 | 86.01 |
| ASH-P | 80.09 | 73.19 | 44.57 | 92.51 | 52.88 | 88.35 | 61.79 | 85.58 | 50.45 | 90.67 | 100.00 | 45.99 | 42.06 | 89.70 | 10.57 | 97.99 | 55.30 | 83.00 |
| ASH-B | 66.08 | 80.33 | 14.21 | 97.32 | 22.08 | 95.10 | 33.45 | 92.31 | 37.63 | 91.97 | 93.05 | 56.82 | 21.17 | 95.50 | 0.06 | 99.58 | 35.97 | 88.62 |
| ASH-S | 64.61 | 81.45 | 11.49 | 97.87 | 27.98 | 94.02 | 39.78 | 90.98 | 32.74 | 92.75 | 89.05 | 67.51 | 11.93 | 97.60 | 0.01 | 99.80 | 34.70 | 90.25 |
| VRA-P | 69.61 | 76.93 | 15.70 | 97.12 | 26.94 | 94.25 | 37.85 | 91.27 | 36.37 | 92.93 | 95.55 | 60.79 | 21.47 | 95.62 | 0.30 | 99.70 | 37.97 | 88.58 |
| Real | 68.91 | 78.58 | 18.64 | 96.53 | 37.76 | 92.69 | 47.31 | 89.97 | 39.63 | 92.33 | 97.45 | 60.01 | 24.15 | 95.57 | 2.90 | 99.34 | 42.09 | 88.13 |
| Gaussian | 69.39 | 78.31 | 18.88 | 96.49 | 37.54 | 92.68 | 47.43 | 89.88 | 39.60 | 92.32 | 97.05 | 60.85 | 23.24 | 95.76 | 3.25 | 99.30 | 42.05 | 88.20 |
| Uniform | 69.34 | 78.49 | 18.83 | 96.51 | 38.71 | 92.51 | 47.98 | 89.81 | 39.51 | 92.46 | 97.70 | 59.49 | 24.24 | 95.54 | 2.58 | 99.37 | 42.36 | 88.02 |
| Dynamic | 68.81 | 78.58 | 18.33 | 96.63 | 36.87 | 92.87 | 45.94 | 90.17 | 39.34 | 92.54 | 97.80 | 59.44 | 24.70 | 95.50 | 2.68 | 99.34 | 41.81 | 88.13 |
| Ours (V) | 68.83 | 78.59 | 18.33 | 96.63 | 37.03 | 92.84 | 45.97 | 90.15 | 39.29 | 92.54 | 97.80 | 59.33 | 24.80 | 95.48 | 2.67 | 99.34 | 41.84 | 88.11 |
| Ours (E) | 68.46 | 78.51 | 15.90 | 97.00 | 34.00 | 93.28 | 43.61 | 90.50 | 37.07 | 92.82 | 96.25 | 60.70 | 21.61 | 95.99 | 0.85 | 99.66 | 39.75 | 88.56 |

Table 9: OOD detection results with ViT-B-16 on ImageNet-1k. ↑ indicates larger values are better and ↓ indicates smaller values are better. All values are percentages averaged over different OOD datasets. **Bold** numbers are superior results whereas underlined numbers denote the second and third best results. "Real" means using a real OOD dataset to resolve Problem (Eq. 10). "Gaussian" and "Uniform" indicate using Gaussian and Uniform distributions as surrogate OOD datasets in resolving Problem (Eq. 10), respectively. "Dynamic" denotes solving a non-convex problem with dynamic weight indices defined in Section B.2.

| Method | Species | | iNaturalist | | SUN | | Places | | OpenImage-O | | ImageNet-O | | Texture | | MNIST | | Average | |
|---|---|---|---|---|---|---|---|---|---|---|---|---|---|---|---|---|---|---|
| | FP↓ | AU↑ | FP↓ | AU↑ | FP↓ | AU↑ | FP↓ | AU↑ | FP↓ | AU↑ | FP↓ | AU↑ | FP↓ | AU↑ | FP↓ | AU↑ | FP↓ | AU↑ |
| MSP | 80.56 | 72.91 | 51.52 | 88.16 | 66.54 | 80.93 | 68.67 | 80.38 | 59.93 | 84.81 | 90.95 | 58.80 | 60.23 | 82.99 | 80.29 | 70.23 | 69.84 | 77.40 |
| ODIN | 82.18 | 67.96 | 52.26 | 85.24 | 66.89 | 76.35 | 69.14 | 75.06 | 58.75 | 81.55 | 89.05 | 54.30 | 56.68 | 81.69 | 79.03 | 58.65 | 69.25 | 72.60 |
| Energy | 86.54 | 61.88 | 64.08 | 79.24 | 72.77 | 70.24 | 74.31 | 68.44 | 64.93 | 76.46 | 87.00 | 52.71 | 58.48 | 79.30 | 83.61 | 52.92 | 73.96 | 67.65 |
| DICE | 88.09 | 65.08 | 90.43 | 74.49 | 94.27 | 65.76 | 92.81 | 65.19 | 82.61 | 77.51 | 85.85 | 70.86 | 83.67 | 77.63 | 99.68 | 74.02 | 89.68 | 71.32 |
| ReAct | 86.82 | 66.08 | 65.15 | 85.98 | 72.46 | 78.97 | 73.74 | 77.51 | 64.71 | 84.19 | 87.30 | 66.70 | 56.95 | 84.65 | 83.41 | 70.81 | 73.82 | 76.86 |
| BFAct | 90.52 | 64.43 | 80.01 | 84.62 | 77.86 | 81.10 | 77.55 | 79.61 | 69.66 | 85.45 | 87.15 | 73.84 | 57.61 | 85.67 | 80.76 | 86.53 | 77.64 | 80.16 |
| ASH-P | 98.38 | 28.99 | 99.95 | 10.42 | 99.69 | 19.83 | 99.66 | 21.02 | 99.57 | 18.91 | 99.15 | 30.70 | 98.49 | 29.92 | 100.00 | 9.57 | 99.36 | 21.17 |
| ASH-B | 94.34 | 53.78 | 92.36 | 48.20 | 95.30 | 52.35 | 95.75 | 52.18 | 92.74 | 52.73 | 94.60 | 51.53 | 93.90 | 43.84 | 99.96 | 18.86 | 94.87 | 46.68 |
| ASH-S | 98.11 | 28.50 | 99.99 | 6.71 | 99.73 | 16.70 | 99.66 | 18.33 | 99.84 | 14.07 | 99.40 | 28.51 | 99.11 | 24.15 | 100.00 | 11.20 | 99.48 | 18.52 |
| VRA-P | 98.14 | 39.19 | 99.78 | 23.06 | 98.83 | 30.67 | 98.89 | 32.59 | 98.28 | 37.46 | 98.65 | 43.67 | 94.57 | 52.22 | 100.00 | 26.40 | 98.39 | 35.66 |
| Real | 85.07 | 67.00 | 64.16 | 88.03 | 70.73 | 83.05 | 71.76 | 81.74 | 65.56 | 87.14 | 89.60 | 73.69 | 63.09 | 84.63 | 79.53 | 87.93 | 73.69 | 81.65 |
| Gaussian | 84.14 | 67.56 | 61.10 | 88.71 | 69.13 | 83.41 | 70.28 | 82.07 | 63.32 | 87.51 | 89.70 | 73.48 | 60.16 | 85.20 | 77.20 | 88.16 | 71.88 | 82.01 |
| Uniform | 84.69 | 67.25 | 62.56 | 88.44 | 70.60 | 83.27 | 71.37 | 81.93 | 64.54 | 87.41 | 90.20 | 73.59 | 61.60 | 84.94 | 79.54 | 87.71 | 73.14 | 81.82 |
| Dynamic | 83.20 | 69.47 | 56.06 | 89.91 | 66.63 | 84.17 | 68.48 | 82.69 | 60.21 | 88.17 | 89.50 | 71.82 | 56.81 | 86.43 | 73.75 | 88.08 | 69.33 | 82.59 |
| Ours (V) | 83.19 | 69.47 | 56.09 | 89.91 | 66.63 | 84.17 | 68.51 | 82.69 | 60.21 | 88.18 | 89.50 | 71.82 | 56.79 | 86.43 | 73.75 | 88.08 | 69.33 | 82.59 |
| Ours (E) | 84.56 | 68.70 | 60.09 | 89.55 | 68.02 | 83.92 | 69.42 | 82.36 | 61.77 | 88.07 | 87.95 | 72.74 | 54.61 | 86.72 | 69.72 | 89.21 | 69.52 | 82.66 |

Table 10: OOD detection results with MLP-B on ImageNet-1k. ↑ indicates larger values are better and ↓ indicates smaller values are better. All values are percentages averaged over different OOD datasets. **Bold** numbers are superior results whereas underlined numbers denote the second and third best results. "Real" means using a real OOD dataset to resolve Problem (Eq. 10). "Gaussian" and "Uniform" indicate using Gaussian and Uniform distributions as surrogate OOD datasets in resolving Problem (Eq. 10), respectively. "Dynamic" denotes solving a non-convex problem with dynamic weight indices defined in Section B.2.

| Method | Species | | iNaturalist | | SUN | | Places | | OpenImage-O | | ImageNet-O | | Texture | | MNIST | | Average | |
|---|---|---|---|---|---|---|---|---|---|---|---|---|---|---|---|---|---|---|
| | FP↓ | AU↑ | FP↓ | AU↑ | FP↓ | AU↑ | FP↓ | AU↑ | FP↓ | AU↑ | FP↓ | AU↑ | FP↓ | AU↑ | FP↓ | AU↑ | FP↓ | AU↑ |
| MSP | 82.15 | 73.21 | 58.99 | 86.84 | 72.98 | 80.92 | 75.52 | 79.83 | 72.42 | 82.59 | 90.95 | 65.84 | 69.95 | 80.27 | 42.36 | 92.95 | 70.66 | 80.31 |
| ODIN | 80.99 | 73.39 | 53.31 | 89.53 | 67.47 | 84.24 | 71.67 | 82.44 | 68.06 | 85.51 | 90.45 | 68.69 | 59.75 | 84.40 | 28.33 | 95.73 | 65.00 | 82.99 |
| Energy | 89.04 | 71.84 | 84.31 | 83.65 | 86.85 | 79.35 | 87.15 | 78.42 | 82.30 | 81.24 | 88.40 | 67.53 | 85.25 | 78.41 | 99.58 | 86.91 | 87.86 | 78.42 |
| DICE | 71.40 | 75.65 | 44.44 | 90.19 | 64.69 | 80.84 | 70.31 | 78.08 | 63.89 | 80.26 | 89.90 | 56.33 | 52.20 | 84.33 | 0.12 | 99.11 | 57.12 | 80.60 |
| ReAct | 91.20 | 70.81 | 91.47 | 80.11 | 91.83 | 77.31 | 90.99 | 76.82 | 85.66 | 79.60 | 87.85 | 67.74 | 87.36 | 77.20 | 94.65 | 87.89 | 90.13 | 77.18 |
| BFAct | 95.94 | 64.38 | 98.60 | 65.76 | 97.16 | 67.50 | 96.66 | 68.74 | 94.69 | 70.00 | 90.85 | 66.57 | 94.61 | 67.83 | 99.85 | 72.19 | 96.04 | 67.87 |
| ASH-P | 97.25 | 46.50 | 99.22 | 32.82 | 98.72 | 40.24 | 97.74 | 43.07 | 98.67 | 38.34 | 94.55 | 52.65 | 99.38 | 31.38 | 100.00 | 1.81 | 98.19 | 35.85 |
| ASH-B | 99.67 | 25.98 | 100.00 | 15.86 | 99.71 | 19.84 | 99.74 | 21.36 | 99.82 | 24.19 | 97.25 | 46.39 | 99.86 | 19.48 | 100.00 | 0.74 | 99.51 | 21.73 |
| ASH-S | 96.50 | 42.89 | 98.42 | 32.35 | 98.03 | 38.00 | 97.90 | 38.63 | 97.65 | 37.03 | 93.50 | 50.19 | 98.90 | 30.41 | 100.00 | 1.80 | 97.61 | 33.91 |
| VRA-P | 98.88 | 29.11 | 99.92 | 11.57 | 99.73 | 17.54 | 99.41 | 20.71 | 99.77 | 16.82 | 97.80 | 37.15 | 99.50 | 16.49 | 100.00 | 0.44 | 99.38 | 18.73 |
| Real | 86.49 | 71.56 | 71.23 | 84.79 | 79.47 | 79.42 | 81.95 | 78.44 | 78.30 | 81.00 | 91.00 | 66.46 | 78.24 | 78.36 | 86.61 | 86.69 | 81.66 | 78.34 |
| Gaussian | 87.00 | 71.10 | 73.67 | 83.35 | 79.82 | 78.80 | 82.07 | 77.86 | 79.06 | 79.83 | 91.20 | 65.95 | 78.90 | 77.64 | 85.01 | 86.10 | 82.09 | 77.58 |
| Uniform | 86.92 | 70.35 | 75.45 | 81.81 | 80.47 | 77.69 | 82.25 | 77.07 | 79.29 | 78.56 | 90.70 | 65.63 | 78.05 | 76.45 | 73.59 | 87.51 | 80.84 | 76.88 |
| Dynamic | 87.24 | 70.88 | 76.95 | 82.17 | 83.83 | 77.26 | 84.75 | 76.12 | 78.65 | 80.31 | 90.70 | 65.68 | 76.76 | 79.17 | 36.06 | 95.19 | 76.87 | 78.35 |
| Ours (V) | 86.98 | 70.98 | 76.40 | 82.36 | 83.45 | 77.42 | 84.67 | 76.27 | 78.35 | 80.46 | 90.85 | 65.77 | 76.40 | 79.26 | 35.66 | 95.19 | 76.59 | 78.46 |
| Ours (E) | 91.67 | 69.17 | 92.48 | 76.83 | 91.83 | 73.88 | 91.86 | 73.34 | 87.14 | 77.34 | 90.40 | 66.02 | 89.29 | 75.33 | 33.61 | 95.61 | 83.54 | 75.94 |

Table 11: OOD detection results on ImageNet-1k where **OOD datasets are iNaturalist, SUN, Places, and Textures**. ↑ indicates larger values are better and ↓ indicates smaller values are better. **Bold** numbers are superior results whereas underlined numbers denote the second and third best results.

| Method | ResNet50 | | MobileNetV2 | | ViT-B-16 | | ViT-L-16 | | SWIN-S | | SWIN-B | | MLP-B | | MLP-L | | Average | |
|---|---|---|---|---|---|---|---|---|---|---|---|---|---|---|---|---|---|---|
| | FP↓ | AU↑ | FP↓ | AU↑ | FP↓ | AU↑ | FP↓ | AU↑ | FP↓ | AU↑ | FP↓ | AU↑ | FP↓ | AU↑ | FP↓ | AU↑ | FP↓ | AU↑ |
| MSP | 64.76 | 82.82 | 70.47 | 80.67 | 61.74 | 83.12 | 65.22 | 81.75 | 59.68 | 83.75 | 62.79 | 81.38 | 69.36 | 81.97 | 76.01 | 80.04 | 66.25 | 81.94 |
| ODIN | 57.68 | 87.08 | 60.42 | 86.39 | 61.24 | 79.59 | 64.06 | 78.65 | 57.30 | 80.99 | 64.36 | 73.77 | 63.05 | 85.15 | 73.17 | 82.29 | 62.66 | 81.74 |
| Energy | 57.47 | 87.05 | 58.87 | 86.59 | 67.41 | 74.31 | 68.43 | 74.65 | 62.82 | 77.65 | 75.32 | 64.87 | 85.89 | 79.96 | 84.44 | 79.38 | 70.08 | 78.06 |
| DICE | 35.72 | 90.92 | 41.93 | 89.60 | 90.30 | 70.77 | 71.77 | 67.12 | 88.68 | 39.90 | 87.92 | 40.71 | 57.91 | 83.36 | 66.23 | 81.53 | 67.55 | 70.49 |
| ReAct | 30.70 | 93.30 | 50.09 | 88.81 | 67.08 | 81.78 | 69.58 | 83.50 | 50.09 | 87.92 | 64.86 | 83.24 | 90.41 | 77.86 | 87.01 | 79.13 | 63.73 | 84.44 |
| BFAct | 31.41 | 92.98 | 48.35 | 89.19 | 73.26 | 82.75 | 81.16 | 82.69 | 57.20 | 88.21 | 65.44 | 86.62 | 96.76 | 67.46 | 96.36 | 72.16 | 68.74 | 82.76 |
| ASH-P | 50.33 | 89.04 | 57.15 | 87.34 | 99.45 | 20.30 | 99.42 | 18.37 | 99.11 | 19.21 | 99.08 | 20.59 | 98.77 | 36.88 | 99.04 | 29.20 | 87.79 | 40.11 |
| ASH-B | 22.73 | 95.06 | 35.66 | 92.13 | 94.33 | 49.14 | 94.08 | 37.89 | 96.42 | 30.00 | 91.47 | 47.38 | 99.83 | 19.14 | 66.05 | 84.31 | 75.07 | 56.88 |
| ASH-S | 22.80 | 95.12 | 38.67 | 90.95 | 99.62 | 16.47 | 99.55 | 16.72 | 99.36 | 14.98 | 99.35 | 17.75 | 98.31 | 34.85 | 99.36 | 18.65 | 82.13 | 38.18 |
| VRA-P | 25.49 | 94.57 | 45.53 | 89.85 | 98.02 | 34.64 | 99.62 | 14.99 | 99.38 | 18.15 | 99.65 | 15.51 | 99.64 | 16.58 | 99.23 | 19.49 | 83.32 | 37.97 |
| Ours (V) | 31.53 | 93.78 | 49.09 | 89.62 | 62.01 | 85.80 | 68.61 | 85.10 | 61.47 | 87.04 | 62.96 | 86.09 | 80.23 | 78.83 | 82.59 | 78.92 | 62.31 | 85.65 |
| Ours (E) | 28.78 | 94.19 | 46.92 | 89.90 | 63.04 | 85.64 | 74.96 | 84.69 | 60.67 | 87.50 | 63.50 | 86.55 | 91.37 | 74.85 | 88.49 | 78.22 | 64.71 | 85.19 |

