# OpenReview forum: "Towards Optimal Feature-Shaping Methods for Out-of-Distribution Detection"
_ICLR.cc/2024/Conference — ICLR 2024 poster_

### Official Review · Reviewer_bZUo · 2023-10-28

**Soundness:** 3 good
**Presentation:** 3 good
**Contribution:** 2 fair
**Rating:** 6
**Confidence:** 4

**Summary:**

The paper finds that the existing feature-shaping methods usually employ rules manually designed for specific model architectures and OOD datasets, which consequently limit their generalization ability. To address this gap, the authors first formulate an abstract optimization framework for studying feature-shaping methods. then they propose a concrete reduction of the framework with a simple piecewise constant shaping function and show that existing feature-shaping methods approximate the optimal solution to the concrete optimization problem. Further, assuming that OOD data is inaccessible, the authors propose a formulation that yields a closed-form solution for the piecewise constant shaping function, utilizing solely the ID data. Through extensive experiments, the authors show that the feature-shaping function optimized by the proposed method improves the generalization ability of OOD detection across a large variety of datasets and model architectures.

**Strengths:**

1. The paper is written well and is easy to understand.
2. The studied problem is very important.
3. The results seem to outperform state-of-the-art.

**Weaknesses:**

1. I am curious about the comparison of the proposed method with the methods that can train with the real outliers in different benchmarks.
2. In terms of the feature-shaping methods, I am curious how well the proposed method compared to the OOD detection using contrastive learning objectives to reshape the feature, such as CSI and SSD [2] on different benchmarks and mode architectures.

[1] CSI: Novelty Detection via Contrastive Learning on Distributionally Shifted Instances

[2] SSD: A Unified Framework for Self-Supervised Outlier Detection

**Questions:**

see above

---

> ### Author Response · Authors · 2023-11-13
>
> We sincerely thank you for the time and effort you invested in reviewing our paper.  We are grateful for your acknowledgment of its well-crafted composition, the significance of the problem it addresses, and its demonstration of superior results compared to current state-of-the-art methods.
>
> We compare our proposed method and others, including training a model with outlier exposure and using contrastive learning objectives. These training methods are not designed for a large-scaled dataset like ImageNet-1k. Thus, our experiments are conducted on the ImageNet-200 benchmark with ResNet18, following OpenOOD V1.5 [1].
>
> We emphasize that this is not a fair comparison, because our method is designed for post-hoc OOD detection, assuming no access to OOD data and no extra model training process. But as shown in the table, there are no significant differences between different methods. Training methods do not show a significant advantage over the post-hoc methods, as also observed by OpenOOD V1.5 [1].
>
> Due to the time constraints associated with the rebuttal, we are currently unable to carry out very extensive experiments to compare post-hoc OOD methods with training methods. However, exploring the performance of training methods across various benchmarks and model architectures is a promising direction that we look forward to investigating.
>
> | Training                    | OOD Score | ID Accuracy                       | AUC (Near OOD) | AUC (Far OOD) | AUC (Mean)   |
> | --------------------------- | --------- | --------------------------------- | -------------- | ------------- | ------------ |
> | **- Post-hoc**              |           |                                   |                |               |              |
> | CrossEntropy                | MSP       | 86.37                             | *83.29*   | 90.13         | 86.71        |
> | CrossEntropy                | Energy    | 86.37                             | 82.41          | 90.84         | 86.62        |
> | CrossEntropy                | ReAct     | 86.37                             | 81.77          | 92.28         | 87.03        |
> | CrossEntropy                | ASH-B     | 86.37                             | 81.60          | *93.77*  | *87.68* |
> | CrossEntropy                | Ours      | 86.37                             | 81.16          | 92.80         | 86.98        |
> | **- Training w/o outliers** |           |                                   |                |               |              |
> | ConfBranch (arXiv’18)       |           | 85.92                             | 79.10          | 90.43         | 84.77        |
> | RotPred (NeurIPS’19)        |           | 86.37                             | 81.59          | 92.56         | *87.08* |
> | G-ODIN (CVPR’20)            |           | 84.56                             | 77.28          | 92.33         | 84.81        |
> | CSI (NeurIPS’20)            |           | Infeasible with compute resources | /              | /             |              |
> | ARPL (TPAMI’21)             |           | 83.95                             | 82.02          | 89.23         | 85.62        |
> | MOS (CVPR’21)               |           | 85.60                             | 69.84          | 80.46         | 75.15        |
> | VOS (ICLR’22)               |           | 86.23                             | 82.51          | 91.00         | 86.75        |
> | LogitNorm (ICML’22)         |           | 86.04                             | 82.66          | *93.04*  | **87.85**    |
> | CIDER (ICLR’23)             |           | No final linear classifier        | 80.58          | 90.66         | 85.62        |
> | NPOS (ICLR’23)              |           | No final linear classifier        | 79.40          | **94.49**     | 86.94        |
> | **- Training w/ outliers**  |           |                                   |                |               |              |
> | OE (NeurIPS’18)             |           | 85.82                             | **84.84**      | 89.02         | 86.93        |
> | MCD (ICCV’19)               |           | 86.12                             | *83.62*   | 88.94         | 86.28        |
> | UDG (ICCV’21)               |           | 68.11                             | 74.30          | 82.09         | 78.19        |
> | MixOE (WACV’23)             |           | 85.71                             | 82.62          | 88.27         | 85.44        |
>
> **Bold** numbers are superior results whereas *Italic* numbers denote the second and third best results.
>
> [1] Zhang, Jingyang, et al. "OpenOOD v1. 5: Enhanced Benchmark for Out-of-Distribution Detection." *arXiv preprint arXiv:2306.09301* (2023).

---

### Official Review · Reviewer_c5SY · 2023-10-30

**Soundness:** 3 good
**Presentation:** 2 fair
**Contribution:** 2 fair
**Rating:** 6
**Confidence:** 3

**Summary:**

This work proposed the feature shaping method in OOD detection. By dividing K intervals, the vector of the K dimension is constructed, named the ISFI vector. Based on this vector, they optimize theta to increase the score of in-distribution and decrease the score of out-distribution. The authors showed the effectiveness of their approach on various architectures compared to existing feature shaping methods.

**Strengths:**

- Compared to previous feature shaping methods, the proposed method is effective on various architectures, especially on MLP architectures.

**Weaknesses:**

- What is the motivation to introduce ISFI? For example, ReAct [A] observes the distribution of per-unit activation for In and out-distribution data.
- I am confused about the meaning of Figure 2. Suppose that z of M dimension is given. Then, theta of ReAct [A] is also M dimension because ReAct conducts the operation on each element of feature z. theta of the proposed method is K dimension determined by the number of intervals. I am not sure whether it is possible to put the plots together.
- theta is optimized over the proposed optimization problem that the closed form exists. However, other feature shaping methods [A, B] do not have the other parameters to be optimized.
- Other feature shaping methods [A, B] said that their methods have comparable performance in distribution distribution after applying their methods. Can authors discuss this point?

[A] React: Out-of-distribution detection with rectified activations, NeurIPS 2021

[B] Extremely Simple Activation Shaping for Out-of-Distribution Detection, 2023

**Questions:**

Please refer to the weaknesses.

==

I raise my score from 3 to 6.

---

> ### Author Response · Authors · 2023-11-13
>
> We greatly appreciate your time and effort in reviewing our paper. We are pleased that you acknowledged the effectiveness of our method across diverse model architectures.
>
> > **Q1: What is the motivation to introduce ISFI?**
>
> We aim to optimize the multiplicative function $\theta(z)$ in Eq. 4, which models elementwise feature-shaping functions. The primary challenge arises from the continuous nature of the feature value $z$, complicating the optimization problem. To address this, we introduce ISFI, a strategic discretization facilitating the assessment of feature importance for OOD detection. Specifically, we partition the domain of features into a disjoint set of intervals, which allows us to approximate the original function $\theta(z)$ with a piece-wise constant function, thus simplifying the optimization problem.
>
> > **Q2: What is the meaning of Figure 2?**
>
> Fig. 2 compares our optimal piecewise constant shaping functions with previous methods, showing that these prior works are empirically approximating our function optimized with access to OOD data.
>
> The methods shown in Fig. 2(a-c), including ReAct, BFAct, and VRA-P, are element-wise shaping functions. These functions identically treat each element of the feature vector, simplifying the comparison.
>
> For example, given any element $z$ of feature representation $\boldsymbol{z}\in \mathbb{R}^M$, our shaping function is $\bar{z}=\theta(z)\cdot z$, where the function $\theta$ is discretized with $K$ parameters. ReAct applies an element-wise shaping function $\bar{z}=\min(z,t)=\frac{\min(z,t)}{z}\cdot z$ to each element. We do not need to compare the two methods on each of the $M$ features, but instead, we compare the two functions $\theta(z)$ and $\frac{\min(z,t)}{z},z\in\mathbb{R}$.
>
> Besides, ASH families in Fig. 2(e-h) incorporate the entire feature vector and are not reducible to element-wise functions. As an approximation, we estimate their average shaping effects across all samples.
>
> > **Q3: Other feature shaping methods such as ReAct and ASH do not have other parameters to be optimized.**
>
> We respectfully argue that this point actually leads to a disadvantage of ReAct and ASH compared with our method. The shaping functions of ReAct and ASH are manually designed for specific model architectures and OOD datasets. This eliminates the need for additional parameters but consequently limit their generalization ability. In contrast, while our shaping function introduces more parameters, it can be efficiently optimized in closed form based on different models and ID data. Additional parameters do not increase user burden; instead, they bring greater flexibility and generalization ability to our method (Tables 1&2).
>
> > **Q4: Can authors discuss the comparable performance claimed by ReAct and ASH?**
>
> There might be some typos in your comment. Here we discuss the generalization ability claimed by ReAct and ASH over different distributions. Please engage with us in discussion if your comment is asking something different.
>
> The experiments conducted in the ReAct and ASH papers involve a narrower range of models and OOD datasets compared to our study. This limitation restricts the evaluation on the generalization ability of their methods.
>
> For instance, when using ImageNet-1k as the ID data, both ReAct and ASH restrict their experiments to two model architectures and four OOD datasets. In contrast, our research encompasses a more extensive evaluation, employing eight different model architectures and eight OOD datasets. Note that the models and OOD datasets used in the ReAct and ASH studies are also included in our experimental setup.
>
> This broader scope in our study allows for a more robust assessment of the generalization ability of our method, significantly contributing to the reliability and applicability of our findings.

---

> ### Comment · Reviewer_c5SY · 2023-11-19
> **Response of comment**
>
> I thank the authors for their response.
>
> > __Q3: Other feature shaping methods such as ReAct and ASH do not have other parameters to be optimized.__
>
> I respect the authors' comments. I agree with the advantage of the proposed method as the authors mentioned. I think that introducing more parameters is a clever approach.
>
> > __Q2: What is the meaning of Figure 2?__
>
> I might have misunderstood the figure at first. I appreciate the authors' comments.
> I have a question about the interpretation of this figure. The authors said that the proposed method flips the sign at low-value features.
> Why do the low-value features are flipped? I think that it is an interesting phenomenon although these parameters are determined by the optimization problem.

---

> > ### Author Response · Authors · 2023-11-19
> >
> > Thank you for your response!
> >
> > > **Why are the low-value features flipped?**
> >
> > We agree with you that it indeed presents a fascinating aspect of feature shaping. Historically, early techniques like ReAct preserve low-value features, whereas subsequent methods such as ASH and VRA-P opt to set these features to zero, achieving improved OOD detection. Our approach instead flips the sign of these features, enhancing their utility in OOD detection.
> >
> > The key to understanding this lies in the relationship between the maximum logit $\ell^{\text{max}}$ and its components: the weight vector $\boldsymbol{w}^{\text{max}}$ and the feature representation $\boldsymbol{z}$,
> >
> > $\ell^{\text{max}}=\sum_{i=1}^{M}w^{\text{max}}_iz_i$.
> >
> > Our experiments revealed that low-value features ($z_i<0.5$) often align with negative weights ($w_i^{\text{max}}<0$). Intriguingly, ID samples generally align these features with smaller weights (on average, $w_{\text{ID}}^{\text{max}}<w_{\text{OOD}}^{\text{max}}<0$). By flipping the sign of these low-value features, we can enhance the maximum logits more significantly for ID samples than for OOD samples, thereby boosting OOD detection performance.
> >
> > It is important to note that Figure 2 is based on a ResNet50 model pre-trained on ImageNet-1k. The optimal element-wise feature-shaping function may vary with other model architectures and datasets. Exploring these variations could be a promising avenue for future research, potentially yielding deeper insights into OOD detection.
> >
> > We are here to assist with any further questions or concerns you may have.

---

> > ### Author Response · Authors · 2023-11-21
> >
> > Dear Reviewer c5SY,
> >
> > Thank you for your insightful feedback on our manuscript. Could you please confirm whether our responses have clarified these points to your satisfaction?
> >
> > We are committed to further enhancing the quality of our work and are readily available to discuss any additional queries or provide further clarifications you might require.
> >
> > Warm regards,
> >
> > The Authors

---

> > > ### Comment · Reviewer_c5SY · 2023-11-21
> > > **Response to Authors**
> > >
> > > I appreciate the authors' comments.
> > >
> > > If there is a visualization to show "low-value features often align with negative weights" directly in the paper, the paper is more convincing. The authors said, "Our experiments revealed...", but it seems to be indirect if the authors refer to Fig. 2. (Or, I would like to ask carefully if it is included in the paper.)
> > >
> > > I wonder whether K-dim ISFI improves the OOD performance.
> > > In other words, $\theta \in \mathcal{R}^m$ can be optimized to $z \in \mathcal{R}^m$  rather than K-dim ISFI.
> > > This experiment can reveal the importance of the optimization problem (Eq. 14) and the interval-specific feature independently. This experiment might show a similar result or tendency in Fig. 4(c) in the main paper.

---

> > > > ### Author Response · Authors · 2023-11-23
> > > >
> > > > Dear Reviewer c5SY,
> > > >
> > > > Thanks for your valuable time and efforts in reviewing our paper! We understand that you may be rather busy during this discussion period.
> > > >
> > > > As the rebuttal stage is drawing to a close within 12 hours, we would like to kindly request your feedback on our responses. We are happy to discuss with you in detail if you have additional comments about our paper. : )
> > > >
> > > > Looking forward to your reply.
> > > >
> > > > Warm regards,
> > > >
> > > > The Authors

---

> > > > > ### Comment · Reviewer_c5SY · 2023-11-23
> > > > > **Response to Authors**
> > > > >
> > > > > I think that the previous methodologies in this field favor simple algorithms [R1, R2, R3, R4, ...].
> > > > > The proposed method provides the closed form of reshaping function not using the out-distribution, and it is not complex. I am considering raising the score more.
> > > > >
> > > > > For enhanced clarity and persuasion, I suggest the following refinements (optional):
> > > > > 1. How about providing the guide of section 3 more specifically? For example,
> > > > >
> > > > >    (1) Start by noting the observation that aligning feature values with negative weights is needed.
> > > > >
> > > > >    (2) Emphasize the difficulty to decide the reshaping function for each architecture or learning algorithm because the characteristics of feature values depends on architectures or learning algorithms.
> > > > >
> > > > >    (3) Also, deciding the reshaping function on all continuous values, which of number is infinite, is difficult.
> > > > >
> > > > >    (4) Propose using piecewise constant over intervals drawing an analogy to visual images are expressed by [0, 255] despite the continuous nature of original colors.
> > > > >
> > > > >    (5) Propose the optimization of the reshaping function over these intervals. As the number of intervals approaches infinity, the method converges to the reshaping function at each value.
> > > > >
> > > > >   **Although the authors do not consider the above refinements (because it is optional), I wonder if the flow of the above is right or not.**
> > > > >
> > > > > 2. I believe that this approach can be extended to other tasks beyond the classification because of the closed form of optimization. If the validity of the method is demonstrated across different tasks compared to existing methods, the paper can show the advantage of the proposed approach.
> > > > >
> > > > >
> > > > > [R1] ReAct: Out-of-distribution Detection With Rectified Activations
> > > > >
> > > > > [R2] Energy-based Out-of-distribution Detection
> > > > >
> > > > > [R3] Extremely Simple Activation Shaping for Out-of-Distribution Detection
> > > > >
> > > > > [R4] VRA Variational Rectified Activation for Out-of-distribution Detection

---

> ### Author Response · Authors · 2023-11-21
>
> Thank you for your response!
>
> > **Visualization to show "low-value features often align with negative weights"**
>
> We apologize for any confusion caused by the phrase "our experiments revealed" in the last reply. The referred experiments are not currently included in our paper. They were designed to empirically explan the specific form of the optimized function, based on a specific model architecture and ID dataset. Note that our optimized function may vary across different model architectures and datasets, so the findings, such as the tendency for low-value features to align with negative weights, might not be universally applicable.
>
> However, in light of your recommendation, we recognize the value of including an additional discussion in the supplementary materials. This will cover the characteristics of the optimal function and provide corresponding empirical explanations. We plan to update the materials accordingly soon.
>
> > **Whether K-dim ISFI improves the OOD performance**
>
> Previous feature-shaping methods predominantly focus on manipulating features based on their values, instead of feature indexes within representations. For instance, ReAct employs truncation for features exceeding a certain threshold, while ASH families disregard features with low values.
>
> Aligned with prior research, our paper seeks to refine the shaping function $\theta$ based on feature values. The introduction of ISFI emerges from the challenges posed by the continuous nature of feature values. By dividing the feature values into separate intervals, ISFI enables us to approximate $\theta(z)$ with a piecewise constant function, thereby simplifying the optimization process. Thus, the primary goal of proposing ISFI is not to enhance OOD performance but to serve as a crucial step in our modeling.

---

> ### Author Response · Authors · 2023-11-21
>
> Dear Reviewer c5SY,
>
> This is a kind reminder that we have updated the supplimentary materials (attached at the end of our manuscript). We added an additional section to discuss the specific form of our optimized shaping function as you suggested. Please refer to Section C where new content have been highlighed in red.
>
> Warm regards,
>
> The Authors

---

> ### Author Response · Authors · 2023-11-23
>
> > **Improve the guide of Section 3**
>
> Thank you for your suggestion to improve the readability of this part. We revise the guide of Section 3. The revised version is as follows:
>
> >> *In this section, we propose a specific instance of the general formulation described in Section 2, following previous element-wise feature-shaping methods (Sun et al., 2021; Xu & Lian, 2023; Kong & Li, 2023). To optimize the shaping function, we initially examine which feature value ranges are effective for OOD detection. However, a significant challenge arises from the continuous nature of feature values. To address this, we partition the domain of features into a disjoint set of intervals, which allows us to approximate the optimal shaping function with a piece-wise constant function. With this approximation, we can formulate an optimization problem to maximize the expected difference between the maximum logit values of ID and OOD data.*
>
> >> *Subsequently, we show that the optimized shaping function using OOD samples bears a striking resemblance to prior methods, such as ReAct (Sun et al., 2021), BFAct (Kong & Li, 2023), and VRA-P (Xu & Lian, 2023). This similarity sheds light on the mechanics of these methods.*
>
> >> *Finally, we propose a formulation that does not require any OOD data to optimize and admits a simple, closed-form solution. Experiments will show its robustness across various datasets and models.*
>
> In your comment, steps 2-5 are basically correct. We did not add step (1) in the guide. As previously mentioned, this step is actually an ad-hoc observation after we have solved our optimization problem, and is specific to a certain model and dataset.
>
> A kind reminder that these editorial changes you are asking for are not related to the technical contribution of our paper. We can improve the readability of the paper further as you, other reviewers and AC suggest in camera-ready version if it is accepted.
>
> > **Show the advantage of the proposed method across different tasks beyond classification**
>
> Different from these earlier studies [R1-4], we extensively expand the range of model architectures and datasets in our experiments, and conduct a thorough comparative analysis between our proposed method and previous ones. The results have shown the improved generalization ability of our method across a large variety of datasets and model architectures.
>
> Previous research [R1-4] and our own work, primarily focus on classification, which is a key and fundamental task in CV. Regarding your suggestion to apply these techniques to other tasks such as object detection and semantic segmentation, it indeed presents an intriguing avenue for exploration. Given the limited time remaining in the discussion period (less than 7 hours), we propose to leave this as a potential future research direction.
>
>
>
> Again, thank you for your taking interest and thinking about how to improve the paper.

---

### Official Review · Reviewer_kmFm · 2023-10-31

**Soundness:** 3 good
**Presentation:** 3 good
**Contribution:** 3 good
**Rating:** 6
**Confidence:** 4

**Summary:**

This paper proposes a feature-shaping method for general OOD detection without OOD data.
This paper uses maximum logit score and energy scores with the manipulated feature to find the OOD data.
This method yields a closed-form solution for the piecewise constant shaping function.
Extensive experiments support the superiority and generalization of this approach.

**Strengths:**

1. The proposed method is generalizable across different benchmarks does not require OOD data, and is somewhat original.
Multiple datasets and multiple backbones are evaluated.
2. This paper is supported by theory as well as experimental validation and is of high quality.
3. The overall description is relatively clear.
4. The OOD task is of great significance. It makes sense to research it.

**Weaknesses:**

1. The reviewer didn't see any discussion or analysis about the robustness of the proposed method to thresholding.
Adding this analysis would make the paper more complete.
2. The reviewer believes that further explanation is needed as to why $E_{x^{OOD} \sim D^{OOD}_{\chi}} [I (Z^{OOD}) ]$ can be ignored.
The explanation in Fig. 3 is still not convincing enough.

**Questions:**

1. Is there a more intuitive explanation for this method that would make it more understandable?
2. Will the code and data be open source?

---

> ### Author Response · Authors · 2023-11-13
> **Official Comment by Authors [1/2]**
>
> We greatly appreciate your time and effort in reviewing our paper. We are pleased that you acknowledged our work's contribution in addressing a significant task, proposing a generalizable method without OOD data, providing strong theoretical and experimental support, and clear writing.
>
> > **Q1: Is there any discussion or analysis about the robustness of the proposed method to thresholding.**
>
> There are two thresholds in the paper. One is the threshold for classifying ID and OOD samples based on OOD scores. Note that AUC already reflects performance robustness across all thresholds. The other pertains to setting feature value limits for optimizing our shaping function, as detailed in Section 3.1. Here we discuss the robustness of our method to the latter.
>
> Given that feature distributions typically exhibit long-tailed characteristics, we select the 0.1 and 99.9 percentiles of all features in a training set as the lower and upper limits of feature values, respectively, to optimize our shaping function. Here we validate different choices of percentiles on the performance of our method, on the ImageNet-1k benchmark with ResNet50. The results are shown as follows:
>
> | Lower percentile | Lower limit | Upper percentile | Upper limit | FPR95     | AUC       |
> | ---------------- | ----------- | ---------------- | ----------- | --------- | --------- |
> | 0.0              | 0.00        | 100.0            | 20.65       | 42.42     | 87.90     |
> | 0.01             | 0.00        | 99.99            | 5.64        | 42.22     | 87.97     |
> | 0.05             | 0.00        | 99.95            | 4.40        | 41.97     | 88.04     |
> | 0.1              | 0.00        | 99.9             | 3.88        | 41.82     | 88.11     |
> | 0.5              | 0.00        | 99.5             | 2.74        | 41.01     | 88.44     |
> | 1.0              | 0.00        | 99.0             | 2.28        | **41.00** | **88.50** |
> | 5.0              | 0.03        | 95.0             | 1.35        | 47.85     | 86.41     |
> | 10.0             | 0.06        | 90.0             | 1.01        | 57.45     | 83.60     |
>
> Our method exhibits a insensitivity to the chooses of lower and upper limits. Although in the main experiments (Tables 1 & 2) we consistently use the 0.1 and 99.9 percentiles, exploring alternate settings reveals potential for slight improvements.
>
> These experiments will be incorporated into the revised paper if space allows or updated in the Supplementary Material.
>
> > **Q2: Further explanation is needed as to why $E_{\boldsymbol{x}^{\text{OOD}}\sim D_{\cal X}^{\text{OOD}}}\left[{\boldsymbol{I}(\boldsymbol{z}^{\text{OOD}})}\right]$ can be ignored.**
>
> In the general setting where we have no prior knowledge about the OOD data encountered during deployment, the distribution of $\boldsymbol{I}(\boldsymbol{z}^{\text{OOD}})$ is unknown, rendering the problem in Eq. 12 intractable. However, if we ignore the unknown term $E_{\boldsymbol{x}^{\text{OOD}}\sim D_{\cal X}^{\text{OOD}}}\left[{\boldsymbol{I}(\boldsymbol{z}^{\text{OOD}})}\right]$, Eq. 12 can be easily solved in closed form and our method shows improved performance for OOD detection. We have some possible working theories on why we can just ignore it (Fig. 3), which are supported by the experiments (Section 4.2).
>
> Our working explanation is based on the similarities or differences in feature representations between these two types of samples.
>
> 1. **In the Case of Hard OOD (High Similarity to ID)**: When OOD samples closely resemble ID samples, their feature representations tend to align. This similarity implies that the expectations we calculate for both OOD and ID samples point in similar directions.
> 2. **In the Case of Easy OOD (Divergence from ID)**: Conversely, when OOD samples are markedly different from ID samples, the maximum logits (a measure of model confidence) for these OOD samples are typically lower. This lower confidence translates to a reduced magnitude of the expectation value for OOD samples compared to ID samples.
>
> Thus, the aspect of $E_{\boldsymbol{x}^{\text{OOD}}\sim D_{\cal X}^{\text{OOD}}}\left[{\boldsymbol{I}(\boldsymbol{z}^{\text{OOD}})}\right]$ that is orthogonal to the ID expectation is minimal, and the term can be ignored with a small affect on the optimized results.
>
> If there are specific concerns or areas that seem unclear, please highlight them and engage with us in the discussion for a more targeted explanation.

---

> ### Author Response · Authors · 2023-11-13
> **Official Comment by Authors [2/2]**
>
> > **Q3: Is there a more intuitive explanation for this method that would make it more understandable?**
>
> Thank you for your question. We write a new explanation, trying to make it easy to understand and planning to revise the paper accordingly. If certain aspects or points are not entirely clear or if you have particular concerns, please point them out for a more focused clarification.
>
> Feature-shaping methods for OOD detection aims to reshape feature representations to better differentiate between ID and OOD samples. Existing methods usually employ shaping functions manually designed for specific model architectures and OOD datasets. In this paper, we want to explore "what feature-shaping function is optimal for OOD detection", focusing on element-wise feature-shaping functions like ReAct. We model this as an optimization problem - optimize the shaping function to maximize an objective function designed for OOD detection. To make the optimization problem tractable, we do two simplifications. The first is to make the shaping function piecewise constant. The second is to use a optimization objective function without OOD terms based on some working theories. The two simplifications lead to a close form solution and both are validated by experiments. Our method improves the generalization ability of OOD detection across a large variety of datasets and model architectures.
>
> > **Q4: Will the code and data be open source?**
>
> Yes. We have included our code as supplementary material alongside this submission. We are also committed to releasing the code on GitHub upon the official acceptance of our paper. Additionally, all the data utilized in our experiments is already open-sourced. To further assist, we will be adding a tutorial in our code repository, detailing the preparation process for the data.

---

### Meta-Review · Area_Chair_RR8z · 2023-12-21

**Metareview:**

This paper presents a feature shaping method for OOD detection, posing it as a closed-form solution to the piecewise constant shaping function. Experiments show the empirical strength of the method. All of the reviewers suggested acceptance, and some of the remaining concerns such as robustness to thresholds, clarification of the method description, and comparison to those that do have outlier exposure. The reviewers participating in the discussion asked several follow-on questions and seemed satisfied with the rebuttal. Since all reviewers recommended acceptance, this paper should be accepted. I encourage the authors to incorporate the elements and results that came out of the discussion.

**Justification For Why Not Higher Score:**

While this paper was positively reviewed, it does not seem like it will have the level of impact and interest from the community to warrant spotlight/oral.

**Justification For Why Not Lower Score:**

Overall, all reviewers recommended acceptance and seemed satisfied with the rebuttal.

---

### Decision · Program_Chairs · 2024-01-16

Accept (poster)